



# Controls on heterotrophic soil respiration and carbon cycling in geochemically distinct African tropical forest soils

Benjamin Bukombe[1], Peter Fiener[1], Alison M. Hoyt[2], Sebastian Doetterl[3,1]

[1]Institute of Geography, Augsburg University, Augsburg, 86159, Germany
[2]Max Planck Institute for Biogeochemistry, Jena, 07745, Germany
[3] Department of Environmental System Science, ETH Zurich, Zurich, 8092, Switzerland

*Correspondence to*: *S. Doetterl (sdoetterl@usys.ethz.ch)*

**Abstract**

Heterotrophic soil respiration is an important component of the global terrestrial carbon (C) cycle, driven by environmental
factors acting from local to continental scales. For tropical Africa, these factors and their interactions remain largely
unknown. Here, using samples collected along strong topographic and geochemical gradients in the East African Rift Valley,
we study how soil chemistry and soil fertility, derived from the geochemical composition of soil parent material, can drive
soil respiration even after many millennia of weathering and soil development.

To address the drivers of soil respiration, we incubated soils from three regions with contrasting geochemistry (mafic, felsic,
and mixed sedimentary) sampled along slope gradients. For three soil depths we measured the potential maximum
heterotrophic respiration under stable environmental conditions as well as the radiocarbon content ($\Delta^{14}C$) of the bulk soil and
respired $CO_2$. We found that soil microbial communities were able to mineralize C from fossil as well as other poor quality
C sources under laboratory conditions representative of tropical topsoils. Furthermore, despite similarities in terms of
climate, vegetation and the size of soil C stocks, soil respiration showed distinct patterns with soil depth and parent material
geochemistry. The topographic origin of our samples was not a main determinant of the observed respiration rates and $\Delta^{14}C$.
In situ, however, soil hydrological conditions likely influence soil C stability by inhibiting decomposition in valley subsoils.
Our study shows that soil fertility conditions are the main determinant of C stability in tropical forest soils. Further, in the
presence of organic carbon sources of poor quality or the presence of strong mineral related C stabilization, microorganisms
tend to discriminate against these sources in favor of more accessible forms of soil organic matter as energy sources,
resulting in a slower rate of C cycling.

Our results demonstrate that even in deeply weathered tropical soils, parent material has a long-lasting effect on soil
chemistry that can influence and control microbial activity, the size of subsoil C stocks, and the turnover of C in soil. Soil
parent material and its lasting control on soil chemistry need to be taken into account to understand and predict C
stabilization and rates of C cycling in tropical forest soils.



## 1. Introduction

### 1.1. Controls on tropical soil C

Tropical forests and the soils therein are one of the most important and largest global terrestrial carbon (C) pools and serve as important climate regulators (Cleveland et al., 2011; Kearsley et al., 2013; Lewis et al., 2009; Sayer et al., 2011a). They contain about one third (421 Pg C) of the global soil organic carbon stock (SOC) in the upper one meter (Köchy et al., 2015) and are characterised by high annual turnover rates (Raich and Schlesinger, 1992). Generally, climate parameters (temperature and precipitation) and vegetation input are regarded as the main factors controlling C dynamics in natural systems (Davidson et al., 2000; Davidson and Janssens, 2006; Rey et al., 2005). Vegetation and climate can stimulate or hamper microbial activity and mineralization of C through quality and quantity of organic matter (OM) input to soil (Fontaine et al., 2007), and the availability of water and energy to drive microbial processes. As a result, modeling SOC dynamics has focused on the influence of parameters related to these two domains to understand C dynamics (Carvalhais et al., 2014; Koven et al., 2017).

However, recent studies show that SOC dynamics are controlled by a much more complex interplay of geochemistry, topography, climate and biology (Doetterl et al., 2015b, 2018; Luo et al., 2017, 2019), much like pedogenesis in general. For example, on average 72% of SOC in humid forest biomes is stabilized by interaction with the mineral phase in soil organo-mineral interactions and occlusion by aggregation (Kramer and Chadwick, 2018). Geology can control C dynamics as soils developed from felsic parent material (high $SiO_2$, low Fe & Al, slow chemical weathering rate) provide less potential for C stabilization and a lower capacity to release rock-derived nutrients than soils developed from mafic parent material (low $SiO_2$, high Fe & Al, fast chemical weathering rate), limiting organic matter input. Additionally, topography through its control on water and soil fluxes may influence C dynamics by altering C respiration and input along slope gradients (Berhe et al., 2008). Hydrological features related to topography in tropical forests are likely to influence C cycling and explain spatial patterns of SOC distribution locally by limiting C decomposition in water saturated valleys (Kwon et al., 2013). Finally, some soils developed from sedimentary parent material can contain a large fraction of fossil organic carbon ($f_{FOC}$) of generally poorer quality than fresh organic matter inputs which can be resistant to decomposition under in situ environmental conditions (Kalks et al., 2020 in review). Hence, in order to explain SOC and its exchange between soil and atmosphere, the interactions of geochemical, geomorphic and climatic drivers are central (Angst et al., 2018; Berhe et al., 2012; Doetterl et al., 2015b; von Fromm et al., 2020 in review; Kramer and Chadwick, 2018; Luo et al., 2017)

To date, it is not clear if the relationships between soil geochemistry, topography and climate identified for temperate ecosystems also apply in the tropics. Especially for the African tropics, more work is required to better understand how soil geochemical, physical, biological and topographic features interact to influence SOC dynamics. Established observatories in African tropical forests focus mostly on biodiversity preservation and C storage in the phytosphere (Tyukavina et al., 2013; Xu et al., 2017), while soils have received much less attention and remain understudied. Generally, data on SOC dynamics



from tropical regions are rare compared to the temperate zone, originating mostly from the Amazon basin (Alberto Quesada
et al., 2020; Schimel et al., 2015; Schimel and Braswell, 2005) and their application to the African tropics may be limited.
For example, atmospheric nitrogen deposition is much higher in sub-Saharan Africa than in other tropical regions due to
large amounts of recurring biomass burning originating from savanna and dry forests north and south of the humid tropics
(Bauters et al., 2018).

## 1.2. Tropical weathering and C dynamics

In many tropical systems, long lasting chemical weathering has led to the depletion of rock-derived nutrients in soils and
limited the capacity of plants to access new nutrients (Vitousek and Chadwick, 2013). It is likely that variation in soil
weathering stage, nutrient availability, as well as the structure and function of tropical forests will affect stabilization
mechanisms and the exchange of C between plants, soil and the atmosphere. Nutrient limitation in highly weathered tropical
soil will likely force plant communities to alter belowground and aboveground C allocation (Doetterl et al., 2015a; Fisher et
al., 2013; Wright et al., 2011), thereby affecting SOC stock. C input to soils is often limited to shallow surface layers, where
roots grow in organic-rich topsoil, with lower C input to deeper soil layers. Further, the stabilization of mineral organic
matter through clay first increases and then decreases with a reduction in reactive mineral surfaces as weathering advances.
In consequence, clay in old tropical soils has a more limited potential to protect C against microbial decomposers compared
to younger soils (Doetterl et al., 2018; Ngongo et al., 2009). In contrast, stable microaggregates rich in iron (Fe) and
aluminum (Al) oxyhydroxides found in abundance in tropical soils (Bruun et al., 2010; Reichenbach et al., 2021 in review)
seem to be of greater importance to stabilize C in tropical soils as concentrations of Al and Fe are commonly higher than in
many temperate soils.

Hence, understanding tropical soil C dynamics ultimately depends on our mechanistic understanding of these complex
interactions and the ability to determine the primary environmental controls on SOC content and respiration. In our study, we
aim to answer if C release through heterotrophic respiration from forest soils in the humid tropics follows predictable
patterns related to geochemical soil properties and topography. We postulate that, in the absence of anthropogenic
disturbance, soil geochemistry derived from its parent material has a lasting effect on soil C respiration due to its influence
on stabilization mechanisms and soil fertility, even in deeply weathered natural tropical soils. Specifically, we study the
factors which control variation in potential heterotrophic soil respiration (SPR) of SOC with varying radiocarbon age ($\Delta^{14}$C)
and stabilized and protected against microbial decomposers by chemical and physical mechanisms. We selected soils in our
study that developed from geochemically distinct parent material along slope gradients under comparable tropical climate
and vegetation. We hypothesize that SPR and the $\Delta^{14}$C signature of SPR in tropical soils are primarily controlled by soil
geochemical properties, driving nutrient availability and C accessibility to microbial decomposers. We further hypothesize



that topography controls SPR and its $\Delta^{14}$C signature indirectly through erosion and deposition processes as well as the hydrological conditions limiting SOC decomposition.

## 2.    Materials and Methods

### 2.1. Study sites

Study sites are located in three forested national parks along the borders of Uganda, Rwanda and the Democratic Republic of
the Congo (DRC), in the East African Rift Valley system. The climate of the study region is classified as tropical humid climate with weak monsoonal dynamics (Köppen Af-Am). Mean annual temperature (MAT) is around 15.3-19.3 °C and mean annual precipitation (MAP) varies between 1697-1924 mm (Fick and Hijmans, 2017). Study sites are located between 1300-2200 m above sea level in sloping mountainous landscapes with small flat plateaus and ridges, followed by longer, steep slopes (up to 60% slope steepness) and small, v-shaped valleys. The dominant vegetation in all forests across the
region is primary tropical mountain forest with smaller differences in biodiversity and species composition (van Breugel et al., 2020; Doetterl et al., 2021 in review; Verhegghen et al., 2012).

Study sites in the DRC are located in Kahuzi-Biega National Park (-2.31439° S; 28.75246° E) where soils have developed from mafic magmatic rocks, a result of volcanism in the East African Rift System (Schlüter, 2006). At our study sites, mafic magmatic rocks are characterized by high Fe and Al and low Si content (Fe: 8.98 ±0.75; Al: 6.26±1.15; Si: 14.22±0.82 (%)
Mean±SE) as well as a high content of rock-derived nutrients such as base cations, calcium (Ca), potassium (K), and Magnesium (Mg) (Ca: 0.58±0.23; K:0.08±0.03; Mg: 1.25±0.13) and phosphorus (P) (0.36±0.05 (%) Mean±SE). Study sites in Uganda are located in Kibale National Park (0.46225° N; 30.37403° E) where soils have developed from felsic magmatic and metamorphic rocks. The felsic magmatic rock in our study region are characterized by the gneissic-granulitic complex with low contents of Fe, and Al, and high Si content (Fe: 1.08±0.5; Al: 0.51±0.38; Si: 37.28±1.87 (%) Mean ±SE). Unlike
mafic, felsic magmatic rocks in our study sites are characterized by low content of rock-derived nutrients such as base cations (Ca: 0.01±0.004; K: 0.01±0.006; Mg: 0.01±0.005) and P (0.005±0.002 (%) Mean±SE). Study sites in Rwanda are located in Nyungwe National Park (-2.463088° S; 29.103834° E) where soils have developed from a mixture of sedimentary rocks of varying geochemistry. These sediments are mostly dominated by quartz-rich sandstones and schist layers spanning along the Congo-Nile divide in the western province of Rwanda (Schlüter, 2006). Similarly, to the felsic magmatic,
sedimentary rocks in our study sites are characterized by low content of Fe and Al but high Si content (Fe: 2.32±0.99; Al: 0.61±0.23; Si: 36.11±4.04 (%) Mean±SE) and low amount of rock derived nutrient including base cations (Ca: 0.005±0.005; K: 0.07±0.03; Mg: 0.01±0.005) and P (0.02±0.009 (%) Mean±SE). A specific feature of the sediment rocks in our study region is the presence of fossil organic carbon of up to 4% C. Fossil organic carbon in these sediments is further characterized by a high C:N ratio (153.9 ± 68.5), depleted in N (Doetterl et al., 2021 in review; Reichenbach et al., 2021 in
review).

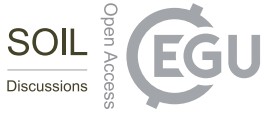

Dominant soil types in the region are various forms of deeply weathered tropical soils. Following the World Reference Base for Soil Resources (IUSS Working Group WRB, 2015), soils in the mafic region are described as umbric, vetic and geric Ferralsol and ferralic vetic Nitisols. Soils in the mixed sediment region and the felsic region are described as geric and vetic Ferralsol. Soils in valley bottoms can locally show gleyic features, and Ferralsols be paired with fluvic Gleysols. Texture

across our sites was generally similar and classified as clay loam with highest clay content in the mafic region (54.20±2.91%), highest silt content in the mixed sediment region (22.63±2.25%) and sand content highest in the felsic region (51.90±1.48%). Deep weathering of the parent material during soil formation is confirmed as bedrock could not be reached in any soils located under forest vegetation (>3 m). The weathering front was found in nearby mining and road cuts only at depths >10 m. Hence, the investigated soils and their geochemical properties, created through many millennia of weathering,

can be interpreted as the end members of pedogenetic alteration for the upper meter of soil, which is the focus of our study.

**2.2. Soil sampling**

As part of a larger project (Project TropSOC) (Doetterl et al., 2021 in review), soil samples were collected following a catena approach with three 40 m x 40 m plots (field replicates) at four topographic positions (plateau, upper slope, midslope and valley/footslope), resulting in 12 plots within each geochemical region (mafic, felsic, mixed sedimentary rocks). Each

plot was subdivided into four subplots of 20 m x 20 m from which four 1meter soil cores were taken using a cylindrical soil corer for undisturbed sampling. The cores were separated into 10 cm increments and the increments from the four cores per plot were mixed into a depth-explicit composite sample. Field moist composite soil samples were subsequently sieved to 12 mm to homogenize the substrate while maintaining aggregate structure. For the experiments conducted in this study, we selected 112 soil samples covering three depth categories including topsoil (0-10 cm), shallow subsoil (30-40 cm), and deep

subsoil (60-70 cm). We selected these three depth intervals as they cover a wide range of biogeochemical properties in soil and various levels of organic matter input to soil, both in terms of quantity (more C input near the surface, less at depth) and quality (Leaf litter + root derived C in topsoil; root-derived C in subsoil). For a more detailed description of the study design, soil sampling, sample treatment and analysis of all biogeochemical parameters used in this study, see the complete database description in Doetterl et al.(2021 in review).

**2.3. Laboratory Experiments**

*Potential heterotrophic soil respiration*

Heterotrophic respiration per gram SOC (specific potential respiration "SPR", $\mu gCO_2$-C $gSOC^{-1}$ $h^{-1}$) and per gram soil (total potential respiration "TPR", $\mu gCO_2$-C $gSoil^{-1}$ $h^{-1}$) was assessed in a lab-based incubation experiment and measured for the three sampling depths across geochemical and topographic gradients. Briefly, 50 g of 12 mm sieved air-dried soil were

weighed into a 100 ml beaker. Soil moisture was adjusted to 60% water holding capacity, selected as the optimum water content level for microbial activity (Rey et al., 2005). Each beaker was placed inside an open 955.5±1.3 ml mason jar covered with parafilm, allowing for air exchange to avoid oversaturation of $CO_2$ within the jar that could inhibit microbial activity. Samples were then incubated at 20 °C, similar to the annual mean temperatures of the study sites. Except for



keeping soil moisture steady by adding water when necessary, no further amendments were made to the incubated soils. Following a pre-incubation period of 4 days to allow equilibration, we incubated all samples for 120 days and sampled periodically every 1 to 14 days throughout the experiment with longer intervals towards the end of the experiment as respiration rates leveled off. The incubation experiment ended when additional $CO_2$ production was not detectable within measurement error. This was the case when the standard deviation of means of the respiration rate between three consecutive measurement time points was smaller than standard deviation between three replicates of the same measurement time point.

For $CO_2$ accumulation prior to sampling, mason jars were sealed for several hours per measurement point. The accumulated $CO_2$ was sampled using a syringe and transferred to pre-evacuated 20 ml vials. To avoid $CO_2$ saturation effects during measurements, potentially influencing microbial decomposition processes, jars were flushed with background air from the laboratory and checked for moisture content, before and after sealing to accumulate $CO_2$. Generally, $CO_2$ samples were taken after accumulating between 1000-3000 ppm $CO_2$. The $CO_2$ concentration of the extracted gas was measured using a gas chromatograph (Trace$^{TM}$ 1300, Thermo Fisher Scientific, Massachusetts, United Stated) calibrated with five $CO_2$ standards covering the range of measured concentrations (0, 500, 1000, 5000, 10000 ppm $CO_2$). Further, the measured $CO_2$ was corrected for the $CO_2$ concentration of the ambient air that was used to flush the jars before closing for $CO_2$ accumulation. After each measurement was completed, each jar was opened and covered with parafilm to allow gas diffusion between $CO_2$ accumulation periods. In this way, an average of 12 observations of $CO_2$ production rate per incubated sample were conducted during the course of the experiment. Data was analyzed as the weighted average of SPR and TPR over the entire length of the experiment with the weight defined by how many days of the incubation experiment each observation represents. We incubated 20% of all samples in triplicate to assess the average difference between samples for the experiment. For these replications, the resulting average standard error of the mean between three lab replicates was 9.6%.

*Δ$^{14}$C of bulk soil and respired $CO_2$*

We measured the soil radiocarbon ($^{14}$C) content of both bulk soil (SOC) and the corresponding respired $CO_2$ from our incubation samples. Radiocarbon analyses were conducted on composite samples of the bulk soil replicates used for incubation and correspondingly on composite samples of the respired $CO_2$ during incubation. Bulk soil Δ$^{14}$C was measured on soil samples before the incubation started. The Δ$^{14}$C of respired $CO_2$ was measured from $CO_2$ that accumulated over the initial period following the pre-incubation period. The $CO_2$ accumulation period varied depending on sample. For top and shallow subsoil with higher $CO_2$ respiration rates it took on average 4-7 days while for deep soil with low $CO_2$ it took 10-15 days to accumulate 1 mg C needed for Δ$^{14}$C analysis. After accumulation, gas samples were collected by attaching a 400 ml evacuated container to the incubation jar using a tube adapter. Every container contained one composite sample resulting from a mixture of the three corresponding replicates.

Radiocarbon concentrations are given as fractions of a modern oxalic acid standard following the conventions of Stuiver and Polach (1977). All measurements were done with the MICADAS Mini Carbon Data System (IonPlus, Switzerland) at the AMS facility at Max Planck Institute for Biogeochemistry (Jena, Germany) (Steinhof et al., 2017).



### 2.4. Assessing fossil vs. biogenic organic carbon

We used radiocarbon measurements to estimate the potential contribution of fossil organic C to the total soil organic C content and to the $CO_2$ respired during incubations. We focused on the mixed sediment region, as the only geochemical region where soil parent material contains fossil organic C. We used a two end member mixing model to calculate the fraction of the C in the sample originating from biogenic vs. fossil organic C as follows:

$$F_{FOC} * f_{FOC} + F_{bio} * f_{bio} = F_{sample} \tag{1}$$

where $F_{FOC}$, $F_{bio}$ and $F_{sample}$ represent the fraction modern radiocarbon content ($F$), of fossil organic C, biogenic C and the measured sample (bulk soil organic C or respired $CO_2$) respectively; and $f_{FOC}$ and $f_{bio}$ represent the proportion of fossil organic C and biogenic C contributing to the sample, respectively. For this estimate, we assumed that fossil organic C is free of [14]C due to the high age of parent material (Doetterl et al., 2021 in review; Schlüter, 2006). Further, we assumed that biogenic SOC in the mixed sediment sites had the same radiocarbon values ($F_{bio}$) with depth as the mean depth-specific radiocarbon content measured from plateau soils of the mafic and felsic regions (regions without fossil organic C) and that these values represent biogenic SOC from active biological cycling in plant-soils systems (Cerri et al., 1985; Kalks et al., 2020 in review). However, because rates of biogenic C cycling likely vary across sites, with potentially slower biogenic cycling in mixed sediment sites (see Discussion), this estimate is likely an upper bound on the fossil organic C contribution to these samples. Based on these assumptions, we reduced Eq. (1) and solved for the proportion of biogenic organic C ($f_{bio}$) as follows:

$$f_{bio} = F_{sample} / F_{bio} \tag{2}$$

The fraction of fossil organic C was then calculated as follows:

$$f_{FOC} = 1 - f_{bio} \tag{3}$$

Finally, since fossil organic C is not renewed, we assessed the time it would take to respire all fossil organic C from soil samples under the conditions of our lab incubation experiments, assuming constant respiration rates, by calculating the ratio of the proportion of fossil organic C ($f_{FOC}$) in the bulk soil, and its respiration rate. The conditions of our incubation experiment, (well aerated as well as sufficient moisture and heat for microbial processes) simulate to some extent the conditions found in topsoils of our investigated tropical forest environment. However, they do not represent in situ conditions for subsoils. Hence, the calculated fossil organic C depletion rates for subsoils would only be applicable to conditions in which subsoil would experience surface environmental conditions, for example, after the erosional removal of topsoil.





## 2.5. Statistical analysis

*Assessing patterns of respiration and $\Delta^{14}C$*

To examine differences in mean SPR and TPR in relation to the three main factors topographic position, soil depth and geochemical region, we conducted three-way analysis of variance (ANOVA). Before ANOVA, we conducted residual

analysis to test for the assumptions of ANOVA, Shapiro-Wilk's test of normality distribution and Levene's test for homogeneity of variances (Shapiro and Wilk, 1965). In most cases, the homogeneity and normality tests did not meet the requirement due to the natural variability of the samples. Hence, we used square root and log transformation to approximately conform to normality and conducted the ANOVA tests on the transformed dataset. To compare the means of multiple groups, post-hoc pairwise comparison was applied using Bonferroni correction (Day and Quinn, 1989) or Tamhane

T2 in the case of unequal variances (Tamhane, 1979).

*Predicting SPR and $\Delta^{14}C$*

Multiple linear regression was used to assess the explanatory power of soil properties to predict SPR and $\Delta^{14}C$ of both bulk soil, respired $CO_2$ and the difference between soil and respired $CO_2$ $^{14}C$ signature ($\Delta$-$\Delta^{14}C$). Before running regression

models, we extracted a wide range of (physico-chemical) soil properties and SOC quality (C:N) indicator for our investigated soils collected by Doetterl et al. (2021 in review), soil C stabilization mechanisms assessed by Reichenbach et al. (2021 in review) and microbial activity parameters (CNP enzymes and microbial biomass), by Kidinda et al. (2020 in review) assessed during our incubation experiment (Table B1). Overall, our dataset consisted of 37 independent variables and 112 aggregated observations for each of our target variables (For $CO_2$: Aggregated from 1350 individual observations on

SPR over the course of the experiment). As multicollinearity and autocorrelation between independent variables was to be expected due to this large number of independent variables and a relatively small number of aggregated observations, we conducted rotated principal component analysis (rPCA) for dimension reduction (Jolliffe, 1995), before regression analysis. We then named all retained rotated components (RCs) based on the loadings of the original variables (Table B1) and interpreted them for the likely underlying mechanisms that can affect C dynamics (Table B1). A threshold of r > 0.5 was

chosen to decide whether an independent variable that is loaded into an RC is used for the mechanistic interpretation of the RC or not. To decide on the number of components to retain for regression analyses, an eigenvalue >1 and explained proportion of variance >5% for each RC were used as criteria to include or exclude RCs into our models (James et al., 2013; Jolliffe, 1995). Furthermore, only RCs were used for predictions when their explanatory power was significant (p<0.1) and clearly distinct from other RCs (evaluated using F-statistics). When predicting our target variables, each regression analysis

was done for three subsets of data: One model containing all data, one with only topsoil data and one with only subsoil data. Samples from valley positions were excluded from this part of the analyses due to the small sample size (9 sites, 27 observations) of the valley subset being too small for reliable regression analyses. Hence, our analyses on identifying controls via regression and rPCS for predicting SPR and $\Delta^{14}C$ is focused on non-valley positions (27 sites, 85 observations).

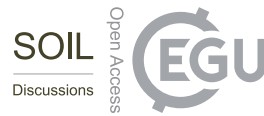

*Assessing relative importance of explanatory variables*

The relative importance of each individual RC in predicting the target variables was assessed by interpreting the effect size of standardized coefficients (-1 to 1) that enter the regression model. When predicting target variables with all data, soil depth was included as an additional explanatory variable beside the rotated components (RCs). We did this in order to avoid interpreting variables as being important for the model when they were instead just auto-correlated to soil depth. In a final step, we used partial correlation analysis following (Doetterl et al., 2015b), to interpret the explanatory power of independent variables in our model, when controlling for soil depth and discuss our findings with respect to microbial (extracellular enzyme activity, see Kidinda et al., 2020 in review), mineralogical (pedogenic oxides, Reichenbach et al., 2021 in review) and soil fertility parameters (available nutrients and exchangeable base cations, see Doetterl et al., 2021 in review) collected for the soils investigated here in other studies. For all statistical tests, due to the relatively small sample size and to avoid Type II statistical errors, a threshold of $p < 0.1$ was used to indicate significant difference. $R^2$ and root mean squared error (RMSE) were used as evaluation metrics for model performance. All statistics were performed using R statistical software and the packages "psych", and "ppcor" (R Core Team, 2020).

## 3. Results

### 3.1 Patterns of respiration and $\Delta^{14}C$

*Topography and soil depth*

We found no statistical difference in specific potential respiration (SPR), total potential respiration (TPR) and radiocarbon content ($\Delta^{14}C$) between plateau and slope positions within each studied geochemical region (mafic, felsic and mixed sediment). Across geochemical regions and soil depths, depth profiles of SPR, TPR and $\Delta^{14}C$ differed only between valleys and non-valley positions (see discussion for details). Within non-valley positions, no statistically significant differences for SPR, TPR and $\Delta^{14}C$ were found between sloping and plateau positions. Hence, all further analyses were done after splitting the data in two subsets: (1) Non-valley positions (plateau, upper slope and midslope) versus (2) valley position (valleys and footslopes) (Fig. 1, 2).

Consistently, for all three geochemical regions in non-valley topographic positions, SPR, TPR (Fig. 1a, c) and $\Delta^{14}C$ (Fig. 2a, c) in soil decreased with soil depth. For SPR and TPR, differences with soil depth were smallest for sites in the mixed sediment and largest for sites in the mafic region. For $\Delta^{14}C$, relative changes with depth were similar for mafic and felsic geochemical regions in both soil and respired $CO_2$, but samples from the mixed sediment region were consistently more depleted in $\Delta^{14}C$ than their counterparts from mafic and felsic regions (Fig. 2).

In valley positions, SPR did not follow a clear trend with soil depth, while TPR did (Fig. 1b, d) decrease with soil depth. For valley positions in the mafic region, SPR decreased with depth, while in the felsic region it increased with depth (Fig. 1b). No statistically significant differences in SPR with depth were observed for the mixed sediment sites (Fig. 1b). All regions show a strong trend of depletion of $\Delta^{14}C$ with depth in valleys (Fig. 2b, d).



*Geochemistry and soil depth*

Consistently, for non-valley profiles, SPR was higher in felsic and mafic regions than in the mixed sediment region (Fig.1a). Additionally, while topsoil samples between mafic and felsic did not show differences in SPR, subsoil samples in the felsic region showed higher SPR than their mafic counterparts and SPR in the mixed sediment region was generally lowest. TPR showed significant differences in topsoil with mafic soils respiring highest and mixed sedimentary rock soils respiring lowest. Differences in subsoil, however, were not significant across regions (Fig. 1C). $\Delta^{14}C$ activity of both soil and respired

$CO_2$ in mafic and felsic regions were not significantly different from each other for both top- and subsoil. Samples from mixed sedimentary rocks were consistently depleted compared to their mafic and felsic counterparts in bulk soil.

At valley positions, SPR in topsoil was not significantly different for mafic and felsic samples. Mixed sediment samples were slightly lower in SPR in topsoil than their mafic and felsic counterparts, but not nearly as low as at non-valley positions (Fig. 1b). In subsoils, SPR was highest in felsic and lowest in the mafic region, with the mixed sediment region being not

significantly different from mafic samples (Fig. 1b). TPR in valleys was significantly different in topsoil with samples from the mafic region respiring highest and from the felsic region lowest. No statistical significant difference of TPR in valleys between geochemical regions could be found in subsoils (Fig. 1d). As in non-valley counterparts, $\Delta^{14}C$ activity was lowest in samples from the mixed sediment region and differences between the mafic and felsic regions were generally small (no statistical test possible due to small sample size, Fig. 2b, d).


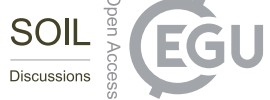

**Figure 1.** Average and standard errors for: specific potential respiration (SPR) as bars (lower panel) and the C:N ratio as points on top of the bars (upper panel), for non-valley positions (**a**) and valley position (**b**); total potential respiration (TPR) as bars (lower panel) and SOC stocks (upper panel) for non-valley (**c**) and valley positions (**d**). (N=9 for non-valleys, and N=3 valleys). Same letters on top of bars indicate no significant difference in SPR or TPR following ANOVA tested for differences between geochemical regions and depth intervals. "x" indicates no significant difference between depth intervals within geochemical region. ANOVA tests were performed separately for non-valley and valley positions.

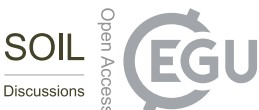

### 3.2 Patterns and differences in $\Delta^{14}C$ soils vs. $\Delta^{14}C$ respired $CO_2$

In general, $\Delta^{14}C$ was highly correlated between bulk soil and respired $CO_2$ ($R^2$ =0.81, p<0.1). In non-valley positions, soil C
was consistently more depleted than its respired C counterparts, and depth trends in the $\Delta^{14}C$ of respired $CO_2$ were much less
pronounced (Fig. 2a, c). Notably, the differences in $\Delta^{14}C$ between soil and $CO_2$ were consistently smaller in the felsic and
mafic regions than in the mixed sediment region. In valley positions, differences in $\Delta^{14}C$ between soils and respired $CO_2$
followed generally the same trends as for non-valley positions, with the exception of $\Delta^{14}C$ of respired $CO_2$ in the mixed
sediment region being similarly depleted as its soil counterpart (Fig. 2b, d).


**Figure 2**. Average and standard errors for (**a**) radiocarbon content ($\Delta^{14}C$) of the respired $CO_2$ for non-valley positions, (**b**)
$\Delta^{14}C$ of the respired $CO_2$ for valley positions, (**c**) $\Delta^{14}C$ of the bulk soil for non-valley positions, and (**d**) $\Delta^{14}C$ of the bulk soil
for the valley positions (n=27 for non-valleys, and n=9 valleys for each depth interval).





We found a significant contribution of fossil organic C to both SOC and respired $CO_2$ in the mixed sediment region (Table 1). There, the calculated contribution of biogenic organic C to total SOC in bulk soil and respired $CO_2$ decreased with soil depth. Similar depth trends were present in valley and non-valley positions. However, the calculated contribution of fossil organic C to respired $CO_2$ was much higher in valley subsoil (19-39% fossil organic C in respired $CO_2$) than in non-valley

subsoil (7-9% fossil organic C in respired $CO_2$). Generally, the contribution of biogenic C to total C was consistently higher (61-97%) in respired $CO_2$ than in SOC of the corresponding bulk soil (48-98%) in both topographic positions. Microbial respiration discriminated against fossil organic C in non-valley positions by a factor of 3-7 ($f_{FOC}$ bulk soil/respired $CO_2$), but did not discriminate against fossil organic C in valley positions (0.7-1.5 $f_{FOC}$ bulk soil/respired $CO_2$). Overall, we note that these values are an upper bound on the contribution of fossil organic C, as these estimates may be impacted by variable rates

of biogenic C cycling (see Discussion). Considering the measured respiration under the conditions of our lab incubation experiments, all fossil organic C in non-valley positions would be mineralized in approximately 447 years from topsoil and in 387-440 years from the subsoil. In valley positions, fossil organic C in topsoil would be mineralized after 61 years. In valley subsoil, fossil organic C would be mineralized after 48-99 years.

**Table 1.** Biogenic and fossil organic carbon contribution in the mixed sediment rock region to SOC and respired $CO_2$ as % of total C and ratio bulk soil / respired C for both parameters. We calculated the time it would take to respire all fossil organic C under the conditions invoked during our lab experiment which simulates topsoil in situ conditions for decomposition. Values are displayed separately for non-valley and valley positions per soil depth. (one observation per position due to merging of replicates into composites prior to analysis).

| Position | Depth [cm] | Biogenic [%] Bulk soil | Respired gas | Bulk/Respired | Fossil [%] Bulk soil | Respired gas | Bulk/Respired | Time to respire all $f_{FOC}$ [Years] |
|---|---|---|---|---|---|---|---|---|
| Non-valley | 0-10 | 89 | 96 | 0.9 | 11 | 4 | 2.8 | 447 |
| | 30-40 | 61 | 93 | 0.6 | 39 | 7 | 6.0 | 440 |
| | 60-70 | 48 | 91 | 0.5 | 52 | 9 | 5.8 | 387 |
| Valley | 0-10 | 98 | 97 | 1.0 | 2 | 3 | 0.7 | 61 |
| | 30-40 | 72 | 81 | 0.9 | 28 | 19 | 1.5 | 48 |
| | 60-70 | 57 | 61 | 0.9 | 43 | 39 | 1.1 | 99 |


### 3.3. Predicting SPR and Δ-Δ$^{14}$C

*Explanatory variables and mechanistic interpretation*

Using data from the non-valley subset, rotated principal component analyses yielded 5 significant rotated components (RCs)
that together explained 74.5% of the cumulative variance of the dataset (Table B1). From these components, RC1 and RC2 explained about 49% of the entire variance in the dataset, and were loaded with 13 (RC1) and 10 (RC2) independent but





highly auto-correlated predictors within each RC. Predictors for RC1 related to soil organic matter characteristics and microbial activity and for RC2 to the chemistry of the soil solution. RC 3-5 explained about 5-11% of the variance within the dataset with varying loading of 2-3 independent predictors that relate mechanistically to texture (RC3), aggregation (RC4)

and C:N ratio + O horizon C stock (RC5).

*Regressions and relative importance of RCs for predicting SPR and Δ-Δ¹⁴C*

Using the rotated components identified above and soil depth as an additional variable, SPR was predicted for the entire non-valley dataset with $R^2 = 0.47$ (RMSE = 1.9 µgCO$_2$-C gSOC$^{-1}$; n = 85). When predicting only topsoils, $R^2$ increased to 0.62

(RMSE = 1.7 µgCO$_2$-C gSOC$^{-1}$; n = 28). When predicting only subsoils, $R^2$ decreased to 0.32 (RMSE = 1.6 µgCO$_2$-C gSOC$^{-1}$; n = 57). Δ-Δ¹⁴C was predicted similarly in all three submodels ($R^2$ = 0.75-0.94; RMSE = 18.1- 88.4 ‰ (Table 2). Besides soil depth, RC1 (soil solution chemistry) and RC3 (texture) were the most important predictors for SPR. Note that in subsoil, texture was no longer picked as a predictor for SPR while it was a highly important predictor in topsoil. Δ-Δ¹⁴C was generally predicted by a wider range of variables than SPR. Interestingly, topsoil Δ-Δ¹⁴C was predicted by the RCs "soil

solution chemistry", "C:N ratio" and "texture". In subsoil, Δ-Δ¹⁴C was predicted by the RCs "SOM and microbial activity", "soil solution chemistry", "aggregation" and "C:N ratio and O horizon C stock". Note that aggregation played only a minor role as predictor in the all data and subsoil prediction of Δ-Δ¹⁴C. Aggregation did not contribute to the prediction power of topsoil Δ-Δ¹⁴C and not in any model for predicting SPR.

**Table 2.** Results of three regression models (all data, topsoil only, and subsoils only) using RCs scores to predict SPR and Δ-Δ¹⁴C including standardized coefficients and model performance indicators. For models using all data, soil depth was included as an additional explanatory variable. Blank cells indicate non significant predictors (*p*-value<0.1) that did not get selected by the model.

| | Standardized Coefficients | | | | | |
| | Topsoil | | Subsoil | | All data | |
| Explanatory variables | SPR | Δ-Δ¹⁴C | SPR | Δ-Δ¹⁴C | SPR | Δ-Δ¹⁴C |
|---|---|---|---|---|---|---|
| Soil depth | | | | | -0.3 | -0.4 |
| SOM and microbial activity (RC1) | | | | 0.4 | | 0.2 |
| Soil solution chemistry (RC2) | 0.5 | 0.4 | 0.4 | 0.5 | 0.5 | |
| Texture (RC3) | -0.7 | -0.4 | | | -0.3 | -0.1 |
| Aggregation (RC4) | | | | -0.3 | | -0.2 |
| C:N ratio and O horizon (RC5) | | -0.6 | | -0.5 | | -0.6 |
| $R^2$ | 0.62 | 0.94 | 0.32 | 0.75 | 0.47 | 0.79 |
| RMSE | 1.7 | 18.1 | 1.6 | 87.4 | 1.9 | 88.4 |
| F-stat | 7.46 | 75.1 | 4.8 | 31.3 | 11.2 | 49.3 |
| *p*-value | 0.0001 | <0.05 | 0.0056 | <0.05 | <0.05 | <0.05 |




*Controlling for soil depth (partial correlations)*

Partial correlation analysis revealed little to no statistical significant changes in correlation between most RCs and our target variables when comparing zero-order and depth-controlled correlations (Fig. 3). However, a marked and significant reduction in correlation was observed between SPR and RC1 (SOM characteristics and microbial activity) as well as between $\Delta$-$\Delta^{14}$C and RC1. A smaller but significant reduction in correlation after introducing soil depth as a control was observed for SPR and RC4 (aggregation). Thus, the reduction in correlation after controlling for soil depth indicates that the relationship of those RCs to target variables is, in parts, depending on soil depth and cannot be interpreted fully independently from it.

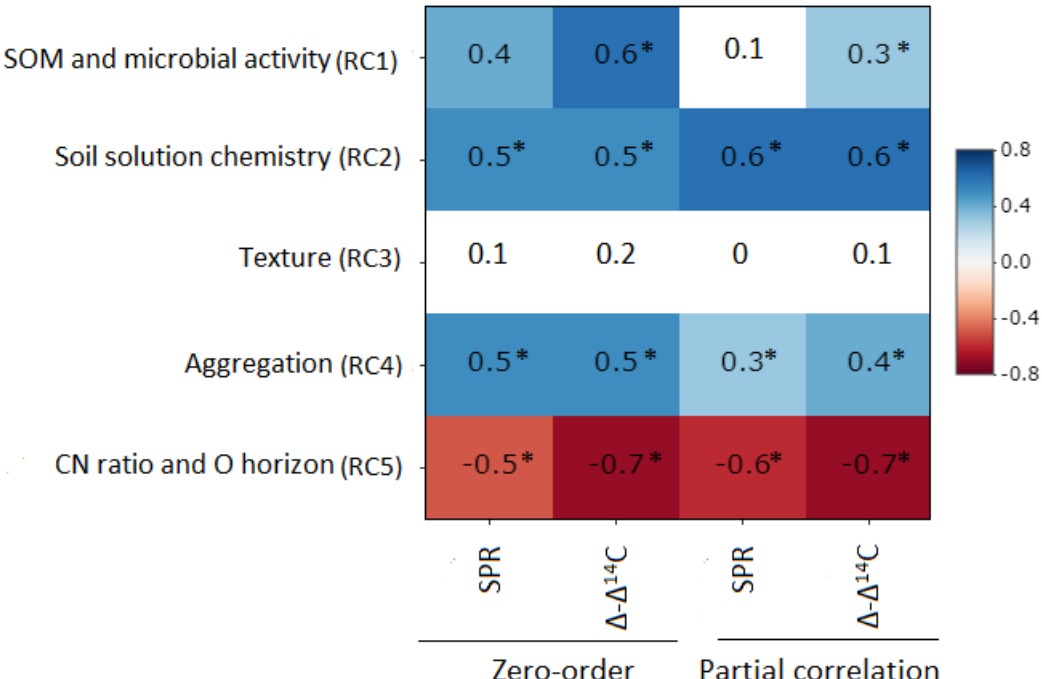

**Figure 3.** Zero order and partial correlation between target variables (SPR and $\Delta$-$\Delta^{14}$C ) and explanatory variables controlled for soil depth. Color indicates the relationship (red = negative correlation and blue = positive correlation, white = correlation was weak, nearly 0). The intensity of the color indicates the strength of the correlation. Asterisks indicate that correlations are significant at *p*-value<0.1.

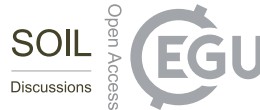

## 4. Discussion

### 4.1. Divergent controls on C dynamics between geochemical regions

*Fertility and microbial activity*

Despite strong similarities in climate and vegetation across sites, the investigated soils showed a remarkable diversity in SPR
and $\Delta^{14}$C. Consistently, the chemistry of the soil solution (RC2) played an important role in predicting SPR and $\Delta^{14}$C in our
lab incubation experiment. This relationship between our target variables and soil solution chemistry was independent of soil
depth (Fig. 3). Additionally, we compared our SPR rates and respired $^{14}$C data with available nutrients (dissolved and
bioavailable C, N and P) and microbial activity (extracellular enzyme activity mining C, N and P) reported by Kidinda et al.
(2020 in review) for the same soils as investigated here and found a positive correlation between SPR and $^{14}$C of respired
$CO_2$ and those two sets of controls (Fig. 4) in the mafic and felsic region, but not in the sediment region.

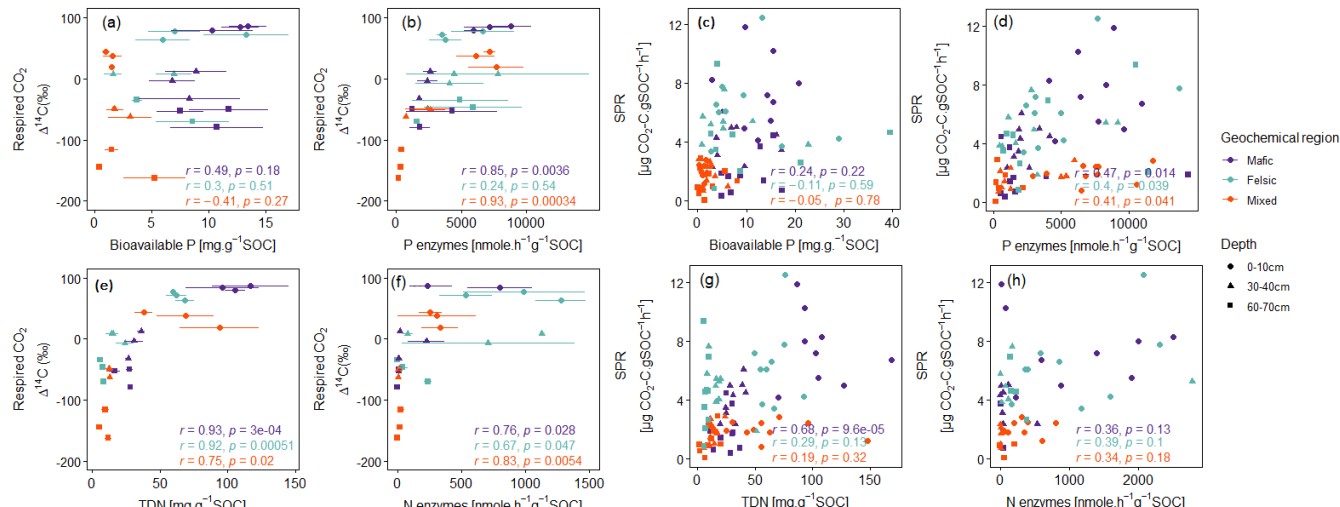

**Figure 4.** Pearson correlation between composite of corresponding replicates of $\Delta^{14}$C of respired $CO_2$ and SPR to P (
panels, a-d), and N (panels, e-h) available nutrient and extracellular enzyme activity data reported by Kidinda et al.
(2020 in review) normalized to SOC content for non-valley positions. Bioavailable P = Bray-P, TDN = Total dissolved
nitrogen. Data displayed in panel a, b, e, and f are averages plus standard errors of three field replicates. Panels c, d, g,
and h show all individual field replicates. Note that two outliers (artifacts) with high bioavailable P values in subsoil
were removed from panels a, and c.

Our data suggests that, in mixed sediment sites, poor soil fertility likely slows down rates of C cycling in soil. Soils in this
region had the lowest available nutrients, with substantially lower concentrations of bioavailable P and N than soils in the
mafic and felsic regions (Fig. 4). While bioavailable P showed a clear distinction between geochemical regions, no strong
linear trends were identified with respect to SPR or the $\Delta^{14}$C of respired $CO_2$ (Fig. 4a, c). In contrast, total dissolved nitrogen
was strongly correlated with SPR and the $\Delta^{14}$C of respired $CO_2$ (Fig. 4b, e) in particular for the mafic and felsic region.



Interestingly, rates of activity for extracellular enzymes mining C, N and P (data from Kidinda et al., 2020 in review) are
similar across all three geochemical regions, but differ between top and subsoil. While the activities of N and P mining
enzymes were positively correlated to SPR (p = 0.01-0.1) in the felsic and mafic sites, we found no significant correlation for
the mixed sediment sites (Fig. 4d-f). No significant correlation with SPR was found for dissolved organic carbon or C
mining enzymes for any sites (data not shown). We interpret this finding as an indication that N & P limitation in our mixed
sediment sites cannot be compensated by microbial decomposer communities. Combined with poor quality fossil organic C,
this leads to reduced respiration rates and older C being respired, both indicators of slower biogenic C cycling. Due to high
fossil C content in the parent material and bulk soil, which also contributes to the respiration of old $CO_2$, we are unable to
quantitatively disentangle the slower biogenic C cycling from the contribution of fossil organic C, using the $^{14}CO_2$ alone.
However, since respiration rates were also significantly lower in topsoil of the mixed sediment sites (Fig. 1), where fossil
organic C content is low (Table 1), soil fertility constraints such as soil exchangeable bases, and bioavailable P (Table B2)
are likely more important for lower respiration rates than the presence of old, potentially more recalcitrant SOC. Thus, our
findings (Table 2) emphasize that the rate at which C is mineralized by microorganisms relates predominantly to soil fertility
conditions and the ability of microorganisms to effectively access nutrients via extracellular enzymes (Kunito et al., 2009;
Schimel et al., 1994), but can be limited by the quality of organic matter sources, or by barriers related to organo-mineral
complexation (Reichenbach et al., 2021 in review).


*The role of tropical weathering in explaining soil respiration*

In contrast to studies on soils in temperate climate zones (Franzluebbers and Arshad, 1997; Hassink, 1997; Schleuß et al.,
2014), aggregation and texture played only a secondary role in explaining variability in SPR and $\Delta^{14}C$, and their influence
decreased with soil depth (Table 2). We explain this observation with the fact that clay minerals at advanced weathering
stages such as kaolinite, dominating in tropical soils, generally show lower activity and reactive surfaces than clay minerals
dominating earlier weathering stages (e.g. Smectite, Vermiculite) (Doetterl et al., 2018). Lower reactivity of these clays and
reduced ability to complex with organic compounds then lead to an overall reduction of the capacity of the clay fraction to
stabilize C in tropical soils compared to temperate soils (Six et al., 2002). Hence, our findings add to the growing body of
literature that demonstrates the limited explanatory power of clay content to explain SOC patterns across large scales (von
Fromm et al., 2020 in review; Rasmussen et al., 2018). Instead, qualitative differences in soil minerals are emerging as an
important predictor for SOC. Furthermore, all subsoils across our study regions are highly weathered and characterized by
low amounts of base cations and low rock-derived nutrient availability (Doetterl et al., 2021 in review; IUSS Working Group
WRB, 2015). Thus, as a second consequence of deep weathering and mineral alterations, remaining nutrients in tropical
forests are cycled predominantly between vegetation, L, O and topsoil horizons where most roots grow (Berish and Ewel,
1988; Cordeiro et al., 2020). Reduced amounts of fresh organic matter input to subsoil and low amounts of rock-derived
nutrients can further increase SOC stability and slow down C turnover (Fig. 2), as these conditions inhibit the ability of soil
microorganisms to decompose more recalcitrant C sources (Fontaine et al., 2007).



*The role of mineral related C stabilization mechanisms*

We observe varying roles of soil C stabilization mechanisms across the investigated geochemical regions, reflected by both the SPR and the $\Delta^{14}C$ signatures in soils. Pyrophosphate extractable organo-mineral complexes (data taken from Reichenbach et al., 2021 in review) were positively correlated to SPR (Fig. A1). In contrast, despite their importance for SOC stocks and for supporting the formation of stable microaggregates that can limit microbial activity, there was no significant zero-order correlation between SPR and Fe, Al- amorphous and crystalline oxides across soils in the investigated

geochemical regions, except for a negative correlation for the mafic region (Fig. A1b, c). We argue that SOC stocks in the mafic region are higher and SPR lower due to the presence of mineral related stabilization mechanisms that are lacking in other regions. Reichenbach et al. (2021 in review) found that higher SOC stocks in the Fe and Al rich mafic region compared to the felsic region are driven by higher amounts of Fe and Al pedogenic oxides that can build stable complexes with organic matter and support the formation of stable microaggregates. These complexes then represent energetic barriers in soil that are

hard to overcome for microorganisms to access potential C resources (Bruun et al., 2010; Zech et al., 1997). Even though the mafic soils were generally more fertile than soils in the felsic or mixed sediment region, SPR was lower and decreased more strongly with depth in mafic soils (75% decrease in deep subsoil compared to topsoil) than in felsic (33% decrease) soils (Fig. 1a) due to the presence of these mineral related stabilization mechanisms.

  In contrast, while SOC stocks in the mixed sediment regions were similar in size to those in the mafic region (Fig. 1),

pedogenic oxide concentrations were similar to those of the felsic region (data not shown, see Reichenbach et al., 2021 in review). Our results indicated that in the mixed sediment region, the presence of fossil organic C and low soil fertility, rather than mineral related stabilization mechanisms, were the main drivers of the high SOC stocks, low SPR and slow rates of C cycling (Fig. 1, Table B2). Thus, SOC stocks, $\Delta^{14}C$ and SPR in our study sites differed, particularly in subsoils, largely due to the geochemical composition of soils (Fig. 1c, d).

Given that annual plant C inputs are high in tropical forest systems (Lewis et al., 2009; Sayer et al., 2011b), exceeding what soils can stabilize, the presence or absence of these stabilization mechanisms is particularly important for long-term soil C stocks. However, $^{14}C$ signatures of bulk soil and respired $CO_2$ for the mafic and felsic sites are similar for the same soil depths and in contrast to lower values in the sediment region (Fig. 2). The presence of fossil organic C in the mixed sediment region results in a distinct difference in the SPR/$^{14}CO_2$ relationship compared to the mafic and felsic region. This shows that

the age of soil C stocks and corresponding transit time of C through the soil between these geochemical regions is driven more strongly by the presence of fossil organic C and a slowing down of C turnover due to the poorer fertility conditions. This interpretation is supported by the fact that SPR and $^{14}CO_2$ were highly correlated with lower SPR corresponding with generally older, more $^{14}C$ depleted SOC at greater depth (Fig. 5). The consistency of this relationship across all three geochemical regions for non-valley positions supports the notion that soil fertility and SOC quality, both lower in subsoils

than in topsoils in our study regions (Table B2, Fig. 5), are the main factors to explain the rate of C cycling in the



investigated tropical soils. However, at valley positions, these relationships are less clear due to the modifying effect of high water saturation on the conservation of potentially labile but old C (Fig. A2, see also detailed discussion 4.2).

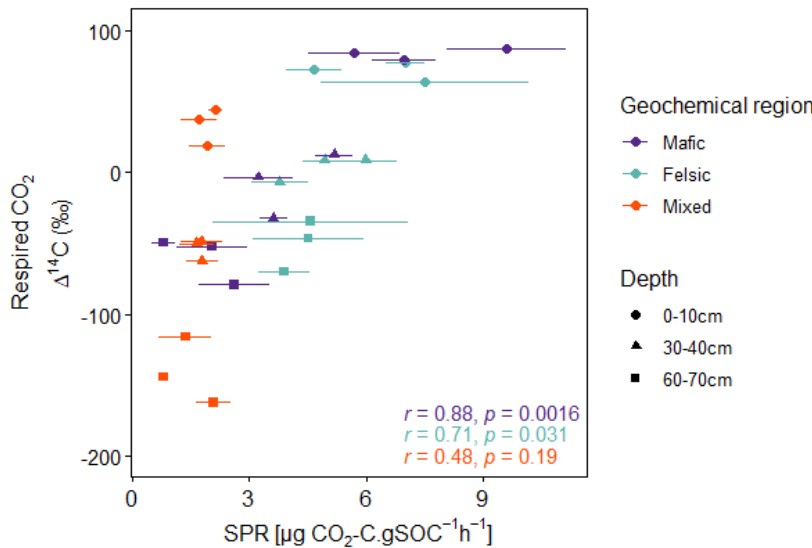

**Figure 5**. Pearson correlation between $^{14}C$ of respired $CO_2$ and SPR for non-valley positions. Data displayed are
averages plus standard error of three field replicates (n=85).

*Accessibility of old C sources to microbial decomposers and its contribution to SOC*

We observed evidence of slower C cycling, the presence of fossil C, and respiration of older C from the mixed sediment region to be higher than in the mafic and felsic regions. Despite similar SOC stocks, SPR and TPR were lowest in soils of
the mixed sediment region, which also had the lowest bulk soil and respired $\Delta^{14}C$ of the three geochemical regions (Fig. 1, 2). This is likely related to the presence of low quality fossil organic C sources, especially in subsoils (Table 1). Fossil organic C at our sites is characterized by C:N ratios (Fig. 1a), largely depleted of nitrogen (Reichenbach et al., 2021 in review), leading to a lower SPR and TPR and decreased microbial activity due to its lower accessibility to decomposers and reduced suitability as a nutrient source (Kalks et al., 2020 in review). The presence of fossil organic C in these soils also had
a marked effect on SOC stocks in subsoils. Without the contribution from fossil organic C, SOC stocks in subsoils of the mixed sediment region would be similarly low to those of the felsic region (Fig. 1c).

There was also evidence that this fossil organic C was microbially available, under certain environmental conditions. Fossil organic C content in soil generally decreased closer to topsoil (Table 1) while large amounts of fossil organic C are still present in subsoil (up to 52% of SOC stock in deeper subsoil). Furthermore, old C sources contributed considerably to
heterotrophic respiration (Fig. 2a, b). Thus, we conclude that under the ideal conditions for microbial activity evoked by our experimental setup, similar to in situ topsoil conditions, microbial organisms can decompose these older, less accessible C



sources (Hemingway et al., 2018). However, the fact that $\Delta^{14}C$ signatures in respired $CO_2$ do not mirror the signature of their C sources in soil indicates that microorganisms do continue to discriminate against these older, poorer C sources if alternatives are available (Fig. 2).

Our data suggests that under the idealized, well-aerated topsoil conditions of our experiment it may take decades to hundreds of years to respire all remaining fossil organic C (Table 1). We explain this observation with the fact that, during laboratory incubations, in the absence of fresh C input, microbes are on the one hand forced to mineralize older and poorly accessible C (in our case: fossil organic C rich subsoil) as an alternative source of energy and nutrients (Feng et al., 2017) thereby decreasing the residence time of the fossil organic C. On the other hand, calculated mineralization rates during our

experiment, particularly for subsoils, are likely higher than in situ. The idealized conditions during our incubation experiment (more oxygen available than in subsoil, ideal moisture and temperature) may have promoted fossil organic C as a feasible C source compared to in situ conditions (Fontaine et al., 2007; Kleber et al., 2005, 2015). On geological timescales, the calculated mineralization rates of fossil organic C are still considerably fast. Being a non-renewable source of organic matter, the fact that fossil organic C can still be found in topsoil is likely related on the one hand to the underlying erosion

rates that continuously degrade the mountainous landscapes of the East African Rift System, and on the other hand to the discrimination against fossil organic C by microbial decomposers in the presence of other, more available C sources. While erosion rates at annual or decadal timescales are negligible for the investigated tropical forests (Drake et al., 2019; Wilken et al., 2020 in review), underlying geological erosion rates estimated for tropical mountain forests globally (Morgan, 2005) range between 0.03-0.2 $t.ha^{-1}.y^{-1}$. Assuming an average bulk density in our study area's topsoil roughly at 1.3 $g.cm^{-3}$

(Doetterl et al., 2021 in review), 6.8-45.3 k years are required to erode the top 10 cm of soil. Thus, slow erosion of soil at millennial timescales may explain the residual content of fossil organic C in topsoil. Even though fossil organic C has the potential to be mineralized under ideal conditions in topsoil (Fig. 2, Table 1), this decomposition and loss of fossil organic C is balanced by the surfacing of former subsoil fossil organic C through topsoil removal.

**4.2 Topographic controls**

*Respiration in tropical forests unaffected by lateral fluxes*

While we did not observe differences in respiration and $\Delta^{14}C$ along slope gradients within any of the geochemical regions, we observe significant differences in SPR between valley and non-valley positions (Fig. 1 and 2). The absence of differences along slopes is a strong indicator that lateral fluxes of matter and water do not significantly influence SOC dynamics. This

finding is supported by work conducted at the global scale where erosion in pristine tropical forests was negligible (Vågen and Winowiecki, 2019). Further, at the regional scale, Drake et al. (2019) found that riverine particulate matter export in rivers draining from pristine tropical forest catchments within our study region are generally dominated by soluble and particulate organic matter fluxes, with little to no mineral sediment being transported. This result is a strong indicator for little to no erosion of mineral soil in pristine catchments, in agreement with our own findings. Nevertheless, over centennial

to millennial time scales, transport of soil material as a result of water movement cannot be ignored in pristine tropical



catchments and greatly influences landscape formation, the long-term rejuvenation of soil surfaces (Flores et al., 2020; Montgomery, 2007) and, in our study, the exposition of fossil C sources to surface conditions. Topography may also play an important role in tropical landscapes on shorter timescales where forests are converted to agricultural use systems, driving erosional processes and sediment export (Berhe et al., 2008; Doetterl et al., 2012; Drake et al., 2019).


*Forest hydrology and soil respiration*

In our study, the effect of topography was limited to differences in hydrological conditions between valleys and non-valley positions. At valley positions decomposition of C in subsoil is generally reduced due to the nearly continuous water saturation, limiting the supply of oxygen (Linn and Doran, 1984; Skopp et al., 1990). These conditions are likely present at

our study sites as supported by our findings of extensive gleyic features in all studied subsoils in valley positions. However, under the ideal conditions for microbial decomposition of C during our laboratory experiment, decomposition of C from valley subsoil was in parts higher than at their non-valley counterparts (Fig. 1). In particular for the sediment region a large proportion of respired C had old $\Delta^{14}$C signatures (Fig. 2). This may be due to the presence of labile C that is inaccessible to microbial decomposers due to unfavourable environmental conditions in valley subsoils. In the laboratory, however, if not

stabilized by other mechanisms such as aggregation or organo-mineral complexation (Reichenbach et al., 2021 in review), these C sources become available to decomposers once environmental constraints, such as water saturation, are removed (Fig. 2b,d). This interpretation is supported by our finding of SPR increasing in valley subsoils of the felsic region (weaker mineral C stabilization through pedogenic oxides) while remaining low in valley subsoils of the mafic (stronger mineral-C stabilization through pedogenic oxides) or the sediment region (poor C quality). (Fig. 1,2; Reichenbach et al., 2021 in

review).

**5. Conclusions**

Our study shows that geochemical differences in the parent material of tropical soils continue to influence the microbial activity, SOC stocks and C turnover even after many millennia of weathering and almost complete pedogenetic alteration of

the parent material. The chemistry of the soil solution, namely soil fertility, and the availability of P and N for microbial decomposers together with C quality were identified as the most important variables explaining patterns of heterotrophic respiration under idealized well-aerated topsoil conditions. C stabilization mechanisms, namely the presence or absence of pedogenic oxides between our geochemical regions were identified as indirect controls to explain variation in soil respiration through their effect on soil aggregation. Patterns of $\Delta^{14}$C with soil depth were largely driven by the presence or absence of

fossil organic carbon of low quality, inherited from parent material. Under idealized well-aerated topsoil conditions, these fossil C sources became available as nutrient sources to microbial decomposers, especially in the absence of better alternative energy and nutrient sources. Furthermore, our analyses revealed that SPR can be driven in parallel by contrasting processes, limiting microbial activity and slowing down C cycling. C in soil of the studied mixed sediment region is low in quality, resulting in low SPR, slower C cycling and high SOC stocks. C in soil of the mafic region is low in accessibility due





to its stabilization with minerals, also leading to low SPR and high SOC stocks. Thus, while the geochemistry differs drastically between soils in those two systems, both show, compared to the felsic region, low SPR for entirely different reasons. While the impact of geochemistry on C dynamics was clearly distinct between the studied soils, topography only played a secondary role in these densely vegetated tropical forest systems. Hydrological features such as water saturation in valleys likely contributed to inhibiting microbial activity in the field, leaving labile C sources available for decomposition

under the idealized laboratory conditions of our experiment. Erosional processes rejuvenate soils and landscapes at geological timescales, but, however, did not account for significant differences in C cycling across our study sites. We conclude from our findings that geochemistry, parent material and its lasting role on pedogenesis are key factors to consider to improve our understanding of C release from tropical forest soils. Improving the spatial representation of C dynamics at larger spatial scales using the variables and controls identified in this study could potentially be an important improvement

for predicting future C turnover and the representation of tropical forest soils in land surface models.

## 6. Appendices

Appendix A-Figures


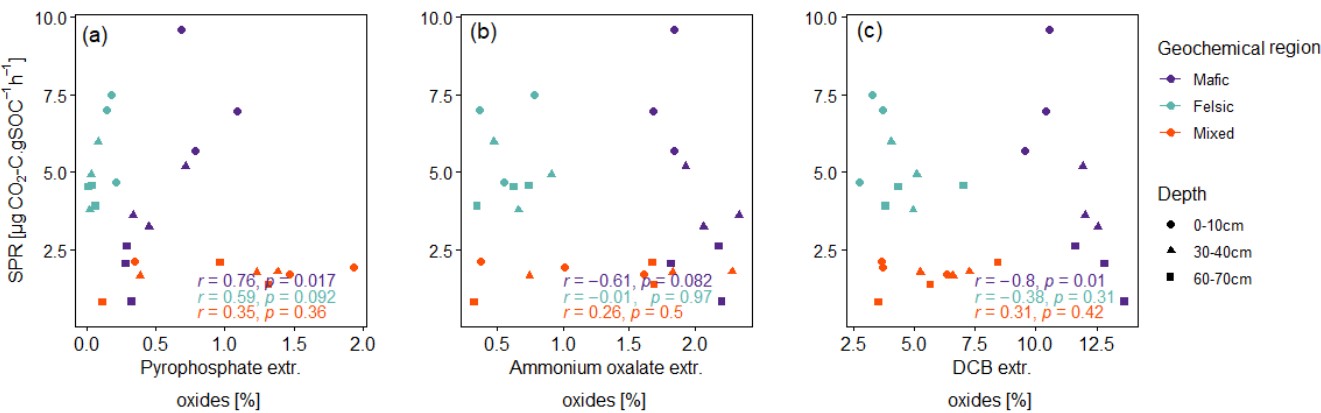

**Figure A1**. Pearson correlation between SPR and sum of pedogenic oxides (Al, Fe and Mn). panel (a) Sodium pyrophosphate extractable oxides, (b) ammonium oxalate-oxalic acid extractable oxides and (c) dithionite-citrate bicarbonate extractable oxides. data reported by Reichenbach et al. (2021 in review).






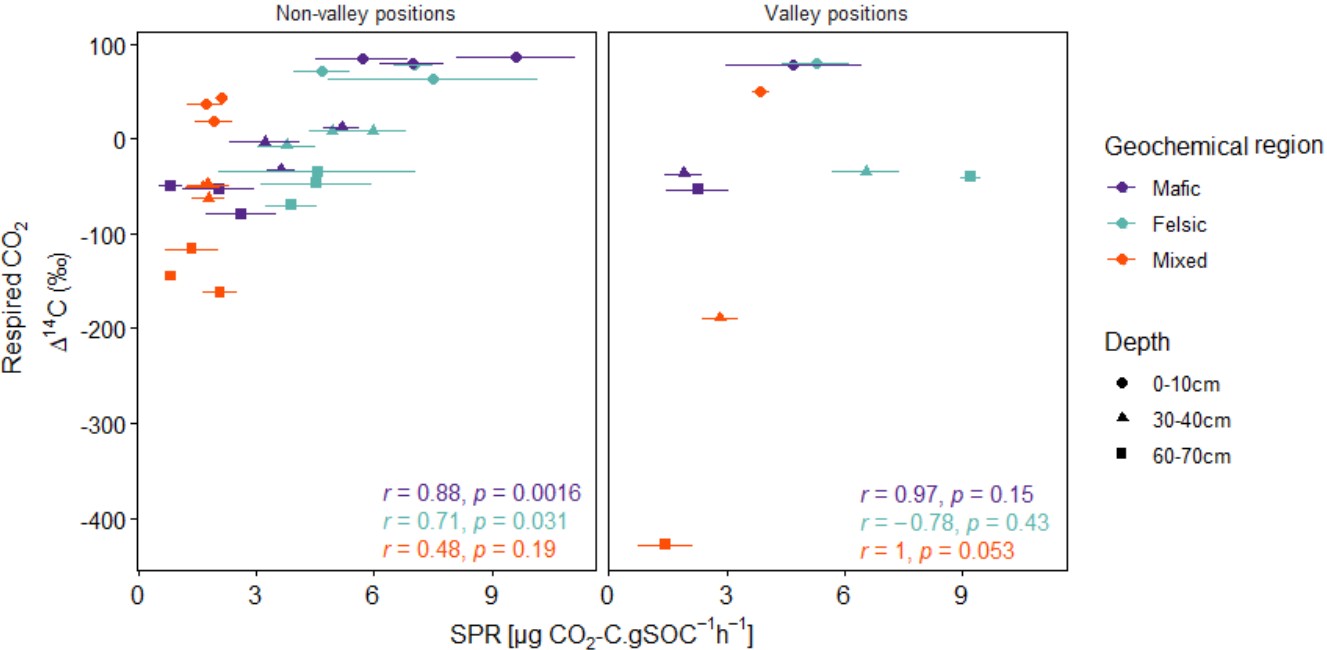

**Figure A2.** Pearson correlation between [14]C of respired $CO_2$ and SPR for non-valley and valley positions. Data displayed are averages plus standard error of three field replicates (non-valleys n=85 and valley position n=27).

Appendix B-Tables

**Table B1.** Rotated principal component analysis for six principal components (RC) retained with Eigenvalues >1 and proportion variance >5%. Upper part of the table shows eigenvalues, individual and cumulative variance and mechanistic interpretation of specific RCs. Bottom part represents loadings with bold marked and underlined values showing the highest loadings of each RC (r>0.5).





| | Rotated component | | RC1 | RC2 | RC3 | RC4 | RC5 |
|---|---|---|---|---|---|---|---|
| | Eigenvalue | | 8.0 | 7.5 | 3.4 | 2.6 | 2.4 |
| | Proportion variance (%) | | 25.0 | 23.6 | 10.5 | 8.0 | 7.5 |
| | Cumulative variance (%) | | 25.0 | 48.6 | 59.1 | 67.1 | 74.5 |
| | Mechanistic interpretation | | SOM and microbial activity | Soil solution chemistry | Texture | Aggregation | C:N ratio and O horizon |
| | Independent variables | Units | | | | | |
| Microbial activity | Carbon enzymes | nmol.g$^{-1}$.h$^{-1}$ | **0.6** | -0.1 | -0.1 | 0.2 | 0.0 |
| | Phosphorus enzymes | nmol.g$^{-1}$.h$^{-1}$ | **0.7** | 0.0 | 0.1 | 0.0 | 0.0 |
| | Nitrogen enzymes | nmol.g$^{-1}$.h$^{-1}$ | **0.6** | 0.4 | 0.0 | 0.1 | 0.0 |
| | Microbial biomass carbon | mg.kg$^{-1}$ | **0.5** | 0.3 | -0.3 | 0.2 | 0.0 |
| Nitrogen | Total dissolved nitrogen | mg.kg$^{-1}$ | **0.9** | 0.1 | 0.1 | 0.1 | 0.0 |
| | Ammonium | mg.kg$^{-1}$ | **0.9** | 0.3 | 0.0 | 0.2 | 0.0 |
| | Nitrate | mg.kg$^{-1}$ | 0.3 | -0.1 | 0.2 | 0.0 | 0.0 |
| | Nitrogen content | % | **1.0** | 0.0 | 0.0 | 0.2 | -0.1 |
| Soil carbon | Dissolved organic carbon | mg.kg$^{-1}$ | **0.8** | -0.3 | -0.1 | 0.0 | -0.1 |
| | Carbon content | % | **0.9** | -0.2 | 0.1 | 0.1 | 0.0 |
| | Soil organic carbon stock | Mg.ha$^{-1}$ | **0.9** | -0.1 | 0.1 | 0.1 | 0.2 |
| | C:N | - | -0.1 | -0.1 | 0.0 | -0.3 | **0.8** |
| C fractions | Microaggregate/silt and clay | % | 0.2 | 0.2 | 0.0 | **0.9** | -0.1 |
| | Relative amount of POM | % | 0.0 | 0.1 | **0.6** | 0.1 | 0.2 |
| | Relative amount of microaggregate | % | 0.2 | 0.1 | -0.3 | **0.8** | -0.3 |
| | Relative amount of silt and clay | % | -0.1 | -0.1 | -0.2 | **-0.8** | 0.1 |
| Soil fertility | Exchangeable acidity | me.100g$^{-1}$ | 0.3 | **-0.9** | 0.0 | 0.0 | 0.0 |
| | Exchangeable bases | me.100g$^{-1}$ | 0.3 | **0.9** | 0.1 | 0.1 | -0.1 |
| | Cations exchange capacity | me.100g$^{-1}$ | **0.7** | -0.2 | -0.4 | 0.2 | -0.3 |
| | Effective cations exchange capacity | me.100g$^{-1}$ | **0.6** | **0.7** | 0.1 | 0.1 | -0.1 |
| | Base saturation in ECEC | % | 0.0 | **0.9** | 0.1 | 0.1 | -0.3 |
| | Base saturation in CEC | % | 0.0 | **1.0** | 0.2 | 0.0 | -0.1 |
| | pH | - | -0.1 | **0.9** | 0.0 | 0.1 | -0.1 |
| | Plant available phosphorus | mg.kg$^{-1}$ | **0.5** | 0.3 | 0.0 | -0.2 | -0.3 |
| Clay activity | pH:Clay | - | 0.0 | **0.5** | **0.8** | 0.0 | 0.0 |
| | Base saturation in ECEC/ Clay | - | 0.0 | **0.9** | 0.3 | 0.1 | -0.2 |
| | Base saturation in CEC/clay | - | 0.0 | **0.9** | 0.3 | 0.0 | -0.1 |
| Texture | Clay | % | -0.1 | -0.2 | **-0.9** | 0.1 | -0.2 |
| | Silt | % | 0.1 | -0.3 | -0.1 | -0.1 | **0.9** |
| | Sand | % | 0.0 | 0.3 | **0.9** | 0.0 | -0.3 |
| C input | O horizon C stock | Mg.ha$^{-1}$ | 0.0 | -0.5 | 0.2 | -0.2 | **0.6** |



**Table B2.** Overview of soil properties and fertility indicators for the three geochemical regions and depth intervals. Abbreviations: Base$_{exc}$= sum of exchangeable bases, CEC = potential cation exchange capacity, ECEC = effective cation exchange capacity; pH$_{KCl}$ =, bio-P = bioavailable phosphorus (Bray P method). Values reported are averages plus standard deviations (n=85). Data reported by Doetterl et al. (2021 in review).

| Geochemical region | Depth [cm] | Base$_{exc}$ [me.100g$^{-1}$] | CEC [me.100g$^{-1}$] | ECEC [me.100g$^{-1}$] | pH$_{KCl}$ | Bio-P [mg.kg$^{-1}$] | TDN [mg.kg$^{-1}$] | Clay [%] | Silt [%] | Sand [%] |
|---|---|---|---|---|---|---|---|---|---|---|
| | 0-10 | 17.2±3.6 | 19.3±3.4 | 18.1±3.8 | 5.33±0.59 | 20.1±13.5 | 141±21.7 | 35±4 | 10±2 | 54±5 |
| Felsic | 30-40 | 6.5±1.5 | 12.3±1.8 | 7.6±1.2 | 4.67±0.69 | 13.8±19.1 | 14.4±7.5 | 41±11 | 8±2 | 51±10 |
| | 60-70 | 5.1±2.2 | 11.2±2.5 | 6.5±1.5 | 4.44±0.72 | 9.6±13.9 | 3.8±1.3 | 49±9 | 7±3 | 44±8 |
| | 0-10 | 7.2±7.3 | 42.6±8.2 | 12.1±4.9 | 3.66±0.60 | 30.8±12.4 | 263.8±100.1 | 54±9 | 14±4 | 33±11 |
| Mafic | 30-40 | 2.6±2.7 | 32.6±2.8 | 7.6±1.3 | 3.6±0.33 | 11.3±6.5 | 44.7±13.9 | 66±7 | 14±4 | 20±4 |
| | 60-70 | 1.1±0.7 | 31.7±4.1 | 6.7±1 | 3.50±0.22 | 11±5.2 | 26.2±5.9 | 67±3 | 13±4 | 20±3 |
| | 0-10 | 0.3±0.1 | 23.1±9.6 | 8.3±1.5 | 3.01±0.20 | 2.9±1.9 | 140.8±87.5 | 36±12 | 20±9. | 44±18 |
| Mixed | 30-40 | 0.2±0.1 | 14.8±5.5 | 4.9±0.9 | 3.57±0.16 | 3.4±2.9 | 20.1±6 | 49±14 | 19±8 | 32±14 |
| | 60-70 | 0.2±0.1 | 12.5±6.7 | 3.8±0.7 | 3.73±0.13 | 2.5±3.19 | 10.6±5.8 | 50±14 | 21±13 | 29±14 |


## 7. Data availability statement

All data used in this study will be published in an open access project-specific database with a separate DOI. The specific data of this publication is available upon request from the corresponding author (S. Doetterl).

## 8. Sample availability

Remaining soil samples are logged and barcoded at the Department of Environmental Science at ETH Zurich, Switzerland.

## 9. Acknowledgements

This study was financed within DFG Emmy Noether group through "Tropical soil organic carbon dynamics along erosional disturbance gradients in relation to soil geochemistry and land use" (TROPSOC; project number 387472333). A.M.H. received funding from the European Research Council (ERC) under the European Union's Horizon 2020 research and innovation program (grant agreement No. 695101 (14Constraint)). The authors would like to thank collaborators of this project: International Institute of Tropical Agriculture (CGIAR-IITA), Max Planck Institute for Biogeochemistry, Institute of Soil Science and Site Ecology at Technical University Dresden, Sustainable Agroecosystems Group and the Soil Resources Group ETH Zurich and the Faculty of Agriculture at the Catholic University of Bukavu, Institut Congolais pour la Conservation de la Nature (ICCN), Rwanda Development Board (RDB) and Uganda Wildlife Authority (UWA). The authors would also like to thank the whole TROPSOC team especially the student assistants for their important work in the laboratory and all field helpers who made the sampling campaign possible. A personal thanks to Mrs Sophie von Fromm for valuable feedback on the manuscript.






## 10. Author contribution statement

S.D., and P.F. designed the research. B.B. conducted the sampling campaign, lab experiments and analyzed the data. S.D.,
P.F., A.M.H. and B.B. interpreted the data. All authors contributed to the writing of the paper.

## 11. Competing interest

SD is a liaison editor of the special issue Tropical biogeochemistry of soils in the Congo Basin and the African Great Lakes
region and PF is a topical editor of the SOIL journal. However, none of them were involved in the review process of this
manuscript. All other authors declare that they have no conflict of interest.

## 12. Special issue statement

This article is part of the SOIL Special Issue: Tropical biogeochemistry of soils in the Congo Basin and the African Great
Lakes region.

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
