# Peer review of "Heterotrophic soil respiration and carbon cycling in geochemically distinct African tropical forest soils"

_SOIL, 2020_

## Author Comment (AC1)

**Reviewer 1**

**Point-by-Point response**

*The article "Controls on heterotrophic soil respiration and carbon cycling in geochemically distinct African tropical forests soils" investigates the role of soil chemistry, fertility and geochemical composition as drivers of soil respiration under laboratory conditions in soils collected along slope gradients in tropical Africa. The article fits the scope of the journal and it will be of great interest for the journal readers.*

**Our response:** We thank the reviewer for the overall positive evaluation, comments, and suggestions. In this response letter, the reviewer's comments are in italic, our responses listed always directly afterward. Suggested text that we will add or remove in the revised manuscript is stated between " " and new/changed text underlined.

Rev 1 Comment 1: *The introduction is to large extents well structured, though I am not sure if the subheaders are really needed.*
**Our response:** Thank you for this recommendation. We agree to remove the two subheaders and connect the two sections.

Rev 1 Comment 2: *The introduction contains a lot of information on geochemical (e.g. Al Fe SiO2) parameters influencing soil C dynamics, but not so much on available P or N, which turn out in this study to be strong determinants of soil respiration.*

**Our response:** We certainly agree. While the role of nutrients on carbon input and stock is briefly introduced in the first paragraph of section 1.2 (lines 71-75), the role of nutrients and microbial activity on soil carbon respiration could be further discussed. We will make the following revisions: We will extend the introduction and add information on nutrients in the revised manuscript, especially between lines 70-72 (see the suggested text underlined below). We will add information related to the work of (Fernández-Martínez et al., 2014; Liu et al., 2015; Jing et al., 2020; Kirsten et al., 2021; and Kallenbach et al., 2016, Mikutta et al., 2019).

"Change in the availability of nutrients such as nitrogen (N) and phosphorus (P) can alter the C cycle as they are tightly coupled to it due to the metabolic needs of plants and microorganisms. Studies have shown that $CO_2$ uptake by terrestrial ecosystems strongly depends on N and P  availability (Fernández-Martínez et al., 2014). Furthermore, N and P can influence microbial growth and activities by changing organic substrate availability and quality (Liu et al., 2015;Jing et al., 2020). Nutrient availability is also driven by rock derived nutrients such as base cations. However, long-term chemical weathering in tropical systems has led to the depletion of rock-derived nutrients in soils and limited the capacity of microorganisms and plants to access new nutrients (Liu et al,2015; Vitousek and Chadwick, 2013). Therefore, variation in soil weathering stage and associated nutrients availability is likely to affect soil C turnover in tropical forests."

**References added:**

Fernández-Martínez, M., Vicca, S., Janssens, I. A., Sardans, J., Luyssaert, S., Campioli, M., Chapin Iii, F. S., Ciais, P., Malhi, Y., Obersteiner, M., Papale, D., Piao, S. L., Reichstein, M., Rodà, F. and Peñuelas, J.: Nutrient availability as the key regulator of global forest carbon balance, Nat. Clim. Chang. |, 4, doi:10.1038/NCLIMATE2177, 2014.

Jing, X., Chen, X., Fang, J., Ji, C., Shen, H., Zheng, C. and Zhu, B.: Soil microbial carbon and nutrient constraints are driven more by climate and soil physicochemical properties than by nutrient addition in forest ecosystems, Soil Biol. Biochem., 141, 107657, doi:10.1016/j.soilbio.2019.107657, 2020.

Liu, L., Gundersen, P., Zhang, W., Zhang, T., Chen, H. and Mo, J.: Effects of nitrogen and phosphorus additions on soil microbial biomass and community structure in two reforested tropical forests, Sci. Rep., 5, doi:10.1038/srep14378, 2015

Rev 1 Comment 3: *Moreover, little information is given in the introduction on the role of aggregation, or microbial biomass as C sequestration 'pump' (or also about microbial enzymatic control). I would recommend extending each of the topics a bit more to make the introduction to make the link even stronger towards the research questions and to the results presented.*

**Our response:** We have introduced the role of aggregation briefly in lines 80-83. But we agree with the reviewer that this could be further strengthened. We propose to add the work of Fang et al. (2017), Rasmussen et al. (2018), Kirsten et al. (2021) and revise this section as follows:

"In contrast, stable microaggregates rich in iron (Fe) and aluminum (Al) oxyhydroxides found in abundance in tropical soils (Bruun et al., 2010; Torres-Sallan et al., 2017) seem to be of greater importance to stabilize C as concentrations of Fe and Al are commonly higher than in many temperate soils. For example, studies conducted across a wide range of tropical ecoregions showed that SOC is mainly regulated by Fe or Al-(hydr) oxides, more so than by clay content. In contrast, clay content is identified in many temperate regions as a major control on SOC (Rasmussen et al., 2018, Fang et al., 2018)".

**References added:**

Torres-Sallan, G., Schulte, R. P. O., Lanigan, G. J., Byrne, K. A., Reidy, B., Simó, I., Six, J. and Creamer, R. E.: Clay illuviation provides a long-term sink for C sequestration in subsoils, Sci. Rep., 7(1), 45635, doi:10.1038/srep45635, 2017.

Rasmussen, C., Heckman, K., Wieder, W. R., Keiluweit, M., Lawrence, C. R., Berhe, A. A., Blankinship, J. C., Crow, S. E., Druhan, J. L., Hicks Pries, C. E., Marin-Spiotta, E., Plante, A. F., Schädel, C., Schimel, J. P., Sierra, C. A., Thompson, A. and Wagai, R.: Beyond clay: towards an improved set of variables for predicting soil organic matter content, Biogeochemistry, 137, 297–306, doi:10.1007/s10533-018-0424-3, 2018.

Fang, K., Qin, S., Chen, L., Zhang, Q. and Yang, Y.: Al/Fe Mineral Controls on Soil Organic Carbon Stock Across Tibetan Alpine Grasslands, J. Geophys. Res. Biogeosciences, 124, 247–259, doi:10.1029/2018JG004782, 2019.

Kirsten, M., Mikutta, R., Vogel, C., Thompson, A., Mueller, C. W., Kimaro, D. N., Bergsma, H. L. T., Feger, K. H. and Kalbitz, K.: Iron oxides and aluminous clays selectively control soil carbon storage and stability in the humid tropics, Sci. Rep., 11(1), 1–12, doi:10.1038/s41598-021-84777-7, 2021.

Rev1 Comment 4: *In addition, I think the hypotheses could be more specifically state, e.g. in line 93 to 96 it would be helpful to mention which change in geochemical properties would cause which response by microbial decomposers more 'explicitly', by stating the expected mechanism, or by hypothesizing under which conditions faster or slower soil C turnover could happen, and how this has been influencing soil C stocks in the long term.*

**Our response:** We will revise the hypotheses following the key points suggested. We propose to change lines 93-96 as follows: "We hypothesize that (1) specific soil respiration and the $\Delta^{14}C$ signature of potential soil respiration in tropical soils are primarily controlled by geochemical properties related to soil fertility (such as potential cation exchange capacity and the sum of base cations) derived and varying with soil parent material. These variations in soil fertility can stimulate or inhibit microbial activity and increase or decrease soil C decomposition rates. (2) The presence of pedogenic oxides can increase SOC stocks and decrease C respiration rate by creating an energetic barrier for C decomposers through direct complexation with organic molecules or by forming stable (micro)aggregates. (3) Topography controls specific soil respiration and its $\Delta^{14}C$ signature indirectly through erosion and deposition processes as well as the hydrological conditions that limit soil microbial growth and activity and slow down soil C decomposition rates."

*Rev 1 Comment 5: The material and methods provide a detailed characterization of the study sites and the respective soil properties including many references to articles that are currently in review, which is a bit difficult to trace. The incubation experiment setup is very clearly described and sound. Also, the statistical analysis is provided in detail, which is great. One minor point that I could suggest to improve the role of soil depth would be to explore linear models and include soil depth 'nested' into topographic position and geochemical region (or nested per sample location), as the different depths are not independent of each other.*

**Our response:** Thank you for this suggestion. We will revise the description of the study sites and present the chemical characteristics in the form of a table as suggested by the second reviewer as well. The text will be revised and shortened. Regarding the statistics and the nested approach, we explored this approach but it did not improve our results. We tested two generalized linear models: one with all factors including geochemical regions, depth intervals, and their interaction; and another one where depth intervals are nested within geochemical regions. We compared the two models using the ANOVA test. The result shows that nesting does not improve the results (see Table below).

Table: Analysis of deviance between Model 1 (where depth intervals are nested within geochemical region) and Model 2 (where we consider interaction of geochemical region and depth intervals). The ANOVA() function performs a test comparing the two models. Here the associated deviance is nearly zero. This provides evidence that model 2 containing the interaction is superior to model 1 with depth interval nested within geochemical region.

```
Analysis of Deviance
Model 1: sqrt(SPR) ~ region/Depth Interval
Model 2: sqrt(SPR) ~ region: Depth Interval
   Resid. Df Resid. Dev Df   Deviance
1       76      6.6006
2       76      6.6006  0 1.7764e-15
```

Rev 1 Comment 6: *The results are well described, some specific suggestions are given below, the figures and tables are adequate. The discussion section is relative to the other parts quite long and reads a bit lengthy. It could maybe be a bit shorter or more streamlined towards the initial research questions and hypotheses. For me, surprising was that in the discussion section further analyses were presented that appear rather as an extension of the results, and may in my opinion therefore also maybe be rather moved to the results section.*

**Our response:** We agree to shorten the discussion section in the revised manuscript. We will do this while keeping the main ideas that are in line with the research questions and hypotheses. We propose the following revision in the discussion part:

" We will remove lines 421-428 (see removed text underlined below) due to their minimal contribution to the discussion. In addition, lines 426-427 provide the same information as in lines 407-410.

While bioavailable P showed a clear distinction between geochemical regions, no strong linear trends were identified with respect to SPR or the $\Delta^{14}C$ of respired $CO_2$ (Fig. 4a, c). In contrast, total dissolved nitrogen was strongly correlated with SPR and the $\Delta^{14}C$ of respired $CO_2$ (Fig. 4b, e) in particular for the mafic and felsic regions. Interestingly, rates of activity for extracellular enzymes mining C, N, and P (data from Kidinda et al., 2020 in review) are similar across all three geochemical regions but differ between top and subsoil. While the activities of N and P mining enzymes were positively correlated to SPR ($p = 0.01$-$0.1$) in the felsic and mafic region, we found no significant correlation for the mixed sediment region (Fig. 4d-f). No significant correlation with SPR was found for dissolved organic carbon or C mining enzymes for any region (data not shown)."

Rev 1 Comment 7: *In addition, it includes the analysis of a much larger set of parameters, which have not really been introduced before (e.g. dissolved organic C, bioavailable P, or enzymes), and I thought, which would already be included in the rPCA analysis? Maybe it would be better to also move the graphs in the appendix section, as they rather support already stated emerging patterns.*

**Our response:** Figure 4 presents variables initially included in the rPCA and represented by the RCs. While the RCs address issues like autocorrelations and model overfitting, they don't provide information on direct relationships between initial variables. In Figure 4 we try to present these relationships masked by RCs. We agree that having this information in the appendices would shorten the discussion section and improve the readability. In the revised manuscript, we will move Figure 4 into the appendix section and revise it to the most essential parts related to the discussion. We believe that these changes will create a direct connection between discussion sections and improve readability.

Rev 1 Comment 8: *In addition, terminology changes also a bit (e.g. mixed region vs. sediment region) – double check, please.*

**Our response:** Thank you for noting this. Based on the elemental composition of the bedrock (see methods), we classified our study area into three geochemical regions including the mafic region, felsic region, and mixed sediment region. We will address any inconsistency throughout the manuscript.

Rev 1 Comment 9: *Moreover, large parts of the discussion are rather discussion the results of the study (see in the technical comments), but there could be more discussion relating the findings to other results found in other tropical/subtropical/montane forest sites and put the results more in to a larger context. Finally, in the discussion a large part is about microbial nutrient limitation, it would be great to already introduce this as a possible control in the introduction in more detail.*

**Our response:** Since this comment is connected to other specific comments (detailed further below), we show briefly how we plan to address each of those comments here. But for details, we direct the reviewer to our specific responses to technical comments related to the discussion. We have planned to make the following changes in the revised manuscript:

1. We will remove lines 474-475; and lines 485-492: as it is overlapping with another paragraph in the "fertility and microbial" section (lines 409-420).
2. We will revise lines 460-468 as detailed in comment 23. In this section, we shortened the discussion on the role of mineral related C stabilization and tried to put it into a larger context in relation to other studies.
3. We also propose to revise lines 497-504 as suggested in comment 24. Here, we screened the text for repeated results and removed them to shorten the discussion.
4. Our response to the question related to microbial nutrient limitation can be found in the response to comment 2. Here, we proposed changes in the introduction section and introduced the nutrient-microbial relationship.

Rev 1 Comment 10: *Line 77: this could need a reference.*

**Our response:** We will support this statement with information from Addo-Danso et al. (2018).

**Reference added:**

Addo-Danso, S. D., Prescott, C. E., Adu-Bredu, S., Duah-Gyamfi, A., Moore, S., Guy, R. D., Forrester, D. I., Owusu-Afriyie, K., Marshall, P. L. and Malhi, Y.: Fine-root exploitation strategies differ in tropical old growth and logged-over forests in Ghana, Biotropica, 50(4), 606–615, doi:10.1111/btp.12556, 2018.

Rev 1 Comment 11: *Line 222: could you provide a reference for subsoil conditions?*

**Our response:** To our knowledge, this is the first study of its kind conducted in our study area on subsoil. Previous studies have mainly reported external environmental conditions. However, previous studies conducted in other tropical regions reported similar results for subsoils. Our statement is supported by the work of Wood et al. (2013) and we will add this information in the revised manuscript.

**Reference added:**

Wood, T. E., Detto, M. and Silver, W. L.: Sensitivity of soil respiration to variability in soil moisture and temperature in a humid tropical forest, PLoS One, 8(12), doi:10.1371/journal.pone.0080965, 2013.

Rev 1 Comment 12: *Line 255: I guess there were no real differences between the plateau, mid-slope, slope positions – still it would be great to mention why these different locations were not considered anymore.*

**Our response:** That is correct, there were no differences between plateau, upper slope, and midslope. As this was a result of statistical analysis, we decided to present this information in the first section of our results in place of the method section (see Lines 277-283 as shown below) and the paragraph below. Since the confusion came earlier in the statistical analysis section, we will therefore move this paragraph back to the statistical analysis section to avoid any confusion. We propose to move the following paragraph between lines 253-254.

"We found no statistical difference in specific potential respiration (SPR), total potential respiration (TPR), and radiocarbon content ($\Delta^{14}C$) between the plateau and slope positions within each studied geochemical region (mafic, felsic, and mixed sediment). Across geochemical regions and soil depths, depth profiles of SPR, TPR, and $\Delta^{14}C$ differed only between valleys and non-valley positions (see discussion for details). For non-valley positions, no statistically significant differences for SPR, TPR, and $\Delta^{14}C$ were found between sloping and plateau positions. Hence, all further analyses were done after splitting the data into two subsets: (1) Non-valley positions (plateau, upper slope, and midslope) versus (2) valley positions (valleys and foot slopes) (Fig. 1, 2)".

Rev 1 Comment 13: *Line 281: The sentence 'Within non-valley positions…' is redundant.*

**Our response:** Thank you for noting this. This is indeed redundant as the levels of non-valley positions are specified at the end of this line. We will revise this sentence as follows: "No statistically significant differences for SPR, TPR, and $\Delta^{14}C$ were found between the plateau and sloping positions."

Rev 1 Comment 14: *Line 299-301: Does this describe exactly the same as is stated in Line 286-288?*

**Our response:** Our aim was to describe the results at the levels of topography and geochemical regions separately. However, the two paragraphs provide indeed the same information. In the revised version, we will exclude lines 299-301(see text below):
"Differences in the subsoil, however, were not significant across regions (Fig. 1C). $\Delta^{14}C$ activity of both soil and respired $CO_2$ in the mafic and felsic regions were not significantly different from each other for both top- and subsoil to avoid repetition."

Rev 1 Comment 15: *Line 315: I don't understand the x indicating no significant difference between depth intervals within geochemical regions, I am not sure which differences the letters demonstrate – within regions or across all regions and depths. Please can you clarify this?*

**Our response:** For the ANOVA analysis, we did the pairwise comparisons between soil depth intervals within geochemical regions and between soil depth intervals across geochemical regions. For Fig.1, x indicates "no significant difference between depth intervals within geochemical region. ANOVA tests were performed separately for non-valley and valley positions."

Rev 1 Comment 16: *Line 324: should there not be two different results? Or should this indicate the Δ14C of bulk soil and of respired CO2 were highly correlated.*

**Our response:** The sentence presents one result " the relationship between $\Delta^{14}C$ of bulk soil and $\Delta^{14}C$ respired $CO_2$". However, we agree that it might be confusing. We will revise this line as follows: We found a strong relationship (high $R^2$) between $\Delta^{14}C$ of the bulk soil and $\Delta^{14}C$ of the respired $CO_2$ ($R^2 = 0.81$, p<0.1).

Rev 1 Comment 17: *Line 393: delete 'from it'.*

**Our response:** Thank you for noting this, we will amend this in the revised version.

Rev 1 Comment 18: *Line 429. Maybe introduce indicators for N & P limitation of microbial decomposers a bit earlier already.*
**Our response:** We will add information in the introduction section, highlighting N and P limitations to microbial decomposers. We will add the following statement before line 429:

"Our data suggest that in mixed sediment region, poor soil fertility likely slows down rates of C cycling in soil. Soils in this region had the lowest available nutrients, with substantially lower concentrations of bioavailable P and N than soils in the mafic and felsic regions (Fig. 4). This adds to existing literature suggesting that nutrient limitation, especially N and P, can significantly slow down microbial growth and activity, hence lowering soil C turnover rates (Fang et al., 2014, Kunito et al., 2009)."

Rev 1 Comment 19: *Line 429: Could you repeat what is considered as poor quality? (e.g. CN ratios of soil organic matter or any other parameters?)*

**Our response:** We used lower nitrogen and higher C:N values to characterize the poor quality of SOC in the mixed sediment region. To clarify this statement, we will revise line 429 as follows: "In addition, the depletion of N and high C:N values ($153.9 \pm 68.5$) of fossil organic C likely contributed to reducing soil respiration rates in the mixed sediment region (Whitaker et al., 2014)."

**Reference added:**

Whitaker, J., Ostle, N., McNamara, N. P., Nottingham, A. T., Stott, A. W., Bardgett, R. D., Salinas, N., Ccahuana, A. J. Q. and Meir, P.: Microbial carbon mineralization in tropical lowland and montane forest soils of Peru, Front. Microbiol. , 5, 720: https://www.frontiersin.org/article/10.3389/fmicb.2014.00720, 2014.

Rev 1 Comment 20: *Line 434: check sentence – lower compared to what – and check tenses – fossil C content 'was' low.*

**Our response:** We will revise this statement as follows: "However, respiration rates in the topsoil of the mixed sediment were also lower compared to the mafic or felsic region (Fig. 1), and fossil organic C content was low compared to the subsoil (Table 1). Thus, we conclude that soil fertility constraints such as soil exchangeable bases, and bioavailable P (Table B2) are likely more important contributors to lower respiration rates in the mixed sediment region than the presence of fossil organic C content."

Rev 1 Comment 21: *Line 439: is there maybe also another study that shows that organo-mineral complexation could be saturated depending on which organo mineral complexes are present in soil (e.g. Quesada 2020, Dötterl 2018).*

**Our response:** We will add new references supporting this statement and revise the sentence as follows: ".....but can be limited by the quality of organic matter sources, or by barriers related to organo-mineral complexation (Rasmussen et al., 2018; Quesada et al., 2020; Traoré et al., 2020; Kirsten et al., 2021)."

Rev 1 Comment 22*: Line 443: add after the brackets: ), in our study aggregation…*

**Our response:** Thank you for noting this, we will revise this sentence.

Rev 1 Comment 23: *Line 460: I would recommend to put this entire section (The role of mineral related C stabilization mechanisms) more into relation with other studies, at the moment, it is rather focusing on either studies from the same data set and reads a bit as an extended results section and could be shortened a bit.*

**Our response:** We agree to revise and shorten this section and put it in a general context in relation to other studies. We will revise "The role of mineral related C stabilization mechanisms" section as follows:

1. We will remove lines 474-475: this does not contribute much to the discussion. In addition, the data discussed here is not presented.
2. We will remove lines 485-492: This paragraph has some overlap with another paragraph in "the fertility and microbial" section (lines 409-420).
3. We will revise lines 460-468 as follows:

*The role of mineral related C stabilization mechanisms*

"Soil C stabilization mechanisms can play an important role in C cycling as reflected by considerable variability of both SPR and $\Delta^{14}C$ signatures in the investigated soils (Fig. A1). For tropical soils, these mechanisms are mainly driven by the abundance of Fe or Al-oxides that improve aggregate stability and ultimately limit microbial activity (Nagy et al., 2018; Kirsten et al., 2021). The importance of pedogenic oxide for SOC stocks was also observed in studies across a variety of ecoregions (Barthès et al., 2008; Rasmussen et al., 2018; Traoré et al., 2020; Quesada et al., 2020). However, despite their importance for SOC stocks, our data did not show a significant relationship between SPR and Fe or Al-oxides across soils in the investigated geochemical regions, except for a negative correlation for the mafic region (Fig. A1b, c). Even though the mafic soils were generally more fertile than soils in the felsic or mixed sediment region, SPR was lower and decreased more strongly with depth in mafic soils (75% decrease in deep subsoil compared to topsoil) than in felsic (33% decrease) soils (Fig. 1a). We argue that SOC stocks in the mafic region are higher and SPR lower due to the presence of mineral related stabilization mechanisms that are lacking in other regions. Consistent with this argument, Reichenbach et al. (2021 in review) found that higher SOC stocks in the mafic region compared to the felsic region are driven by higher amounts of Fe and Al pedogenic oxides that can build stable complexes with organic matter and support the formation of stable microaggregates."

**References added:**

Nagy, R. C., Porder, S., Brando, P., Davidson, E. A., Figueira, A. M. e. S., Neill, C., Riskin, S. and Trumbore, S.: Soil Carbon Dynamics in Soybean Cropland and Forests in Mato Grosso, Brazil, J. Geophys. Res. Biogeosciences, 123, 18–31, doi:10.1002/2017JG004269, 2018.

Rasmussen, C., Heckman, K., Wieder, W. R., Keiluweit, M., Lawrence, C. R., Berhe, A. A., Blankinship, J. C., Crow, S. E., Druhan, J. L., Hicks Pries, C. E., Marin-Spiotta, E., Plante, A. F., Schädel, C., Schimel, J. P., Sierra, C. A., Thompson, A. and Wagai, R.: Beyond clay: towards an improved set of variables for predicting soil organic matter content, Biogeochemistry, 137, 297–306, doi:10.1007/s10533-018-0424-3, 2018.

Traoré, S., Thiombiano, L., André Bationo, B., Kögel-Knabner, I. and Wiesmeier, M.: Organic carbon fractional distribution and saturation in tropical soils of West African savannas with contrasting mineral composition, doi:10.1016/j.catena.2020.104550, 2020.

Quesada, C. A., Paz, C., Oblitas Mendoza, E., Phillips, O. L., Saiz, G., and Lloyd, J.: Variations in soil chemical and physical properties explain basin-wide Amazon forest soil carbon concentrations, SOIL, 6, 53–88, https://doi.org/10.5194/soil-6-53-2020, 2020.

Kirsten, M., Mikutta, R., Vogel, C., Thompson, A., Mueller, C. W., Kimaro, D. N., Bergsma, H. L. T., Feger, K. H. and Kalbitz, K.: Iron oxides and aluminous clays selectively control soil carbon storage and stability in the humid tropics, Sci. Rep., 11(1), 1–12, doi:10.1038/s41598-021-84777-7, 2021.

Rev 1 Comment 24: *Line 506 the same suggestion as above, I think this section can be shortened too, screen for repeated results.*

**Our response:** Thank you for this comment. We will address this in the revised version of the manuscript. We propose the following changes in this section:

1. We will remove lines 497:504 since this paragraph discusses ideas that are already covered in the previous sections.
2. We will then revise lines 497-504 as follows:

"The presence of fossil organic C in the mixed sediment region (up to 52% of SOC stock in deeper subsoil) (Table 1), had a marked effect on SOC stocks in subsoils that would otherwise be similarly low to those of the felsic region (Fig. 1c). Consistent with this finding, a recent study shows that fossil organic C can largely contribute to SOC in subsoils (Kalk et al., 2020, in review). While fossil organic C in our study region is of poor quality as indicated by high C:N ratio values and depleted N, our data shows that fossil organic C was microbially available, under certain environmental conditions (i.e. topsoil conditions (Fig. 2))."

Rev 1 Comment 25: *Line 538: check tenses – there is sometimes a switch between present and past tense within sentences.*

**Our response:** Thank you for noting this. "Observe" should be "observed". We will correct this sentence and check for similar inconsistencies throughout the manuscript.

Rev 1 Comment 26: *Line 570: namely twice in the same paragraph*

**Our response:** We will delete the second (line 572) and revise that line as follows: C stabilization mechanisms, including the presence or absence of pedogenic oxides between our geochemical regions, were identified as indirect controls to explain variation in soil respiration through their effect on soil aggregation.

---

## Author Comment (AC2)

**Reviewer 2**

**Point-by-Point response**

*In this manuscript, the authors present the results of a 120 day soil incubation study of three soil types (mafic, felsic, and mixed sediment) sampled across topographic positions and investigate the biological and chemical controls on respiration rates and respired 14CO2. This manuscript is a significant contribution to the soil organic carbon research because the soils are from tropical Africa, a region where few studies have been carried out. Furthermore, this manuscript looks at the important C stabilization mechanisms in highly weathered soils, which are also poorly sampled and understood relative to less weathered soils often found in temperate regions. Another positive is that the researchers examined 3 depths within the soil profile. I found no glaring issues with the methods used in this study. I did have some questions about the way some of the data were presented. On one hand, this paper could be shorter and more streamlined, but on the other hand the lack of data from Tropical Africa does greatly increase the value of all the parameters reported here. I offer some suggestions on how to shorten the paper below.*

**Our response:** We thank the reviewer for the overall positive evaluation, comments, and suggestions. In this response letter, the reviewer's comments are in italic, our responses listed always directly afterward. Suggested text that we will add or remove in the revised manuscript is stated between " " and new/changed text underlined.

Rev 2 Comment 1: *Lines 107-125: This whole paragraph would be better as a Table introducing the sites and their location and chemistry. A table would be easier to read than a paragraph and make it easier to compare the sites.*

**Our response:** We agree that this section can be shortened by adding a summary table. We will summarize the chemical composition of the sites in the form of a table (see table below) and revise lines 107-125 as described below.

"Study sites in the Democratic Republic of the Congo are located in Kahuzi-Biega National Park (-2.31439° S; 28.75246° E) where soils have developed from mafic magmatic rocks, a result of the active volcanism in the East African Rift System (Schlüter, 2006). Mafic magmatic rocks in the region are characterized by high Fe and Al and low Si content as well as a high content of rock-derived nutrients such as base cations, and phosphorus (P) (Table 1). Study sites in Uganda are located in Kibale National Park (0.46225° N; 30.37403° E) where soils have developed from felsic magmatic and metamorphic rocks. The felsic magmatic rocks in our study region are characterized by the gneissic-granulitic complex with low contents of Fe, and Al, and high Si content. Unlike mafic, felsic magmatic rocks in our study sites are characterized by low content of rock-derived nutrients (Table 1). Study sites in Rwanda are located in Nyungwe National Park (-2.463088° S; 29.103834° E) where soils have developed from a mixture of sedimentary rocks of varying geochemistry. These sediments are mostly dominated by quartz-rich sandstones and schist layers spanning along the Congo-Nile divide in the western province of Rwanda (Schlüter, 2006). Similarly to felsic magmatic rocks, sedimentary rocks in our study sites are characterized by low content of Fe and Al but high Si content and low amount of rock derived nutrients (Table 1). A specific feature of the sediment rocks in our study region is the presence of fossil organic carbon of up to 4% C. Fossil organic carbon in these sediments is further characterized by a high C:N ratio (153.9 ± 68.5), and depleted in N (Doetterl et al., 2021 in review; Reichenbach et al., 2021 in review)."

Table: Chemical composition of the unweathered rock samples representing the soil parent material in three geochemical regions. Values represent mean± standard errors.

| Geochemical region | C[%] | Fe [%] | Al [%] | Si [%] | Ca [%] | K [%] | Mg [%] | P [%] |
|---|---|---|---|---|---|---|---|---|
| Mafic | 0 | 8.98 ±0.75 | 6.26±1.15 | 14.22±0.82 | 0.58±0.23 | 0.08±0.03 | 1.25±0.13 | 0.36±0.05 |
| Felic | 0 | 1.08±0.5 | 0.51±0.38 | 37.28±1.87 | 0.01±0.004 | 0.01±0.006 | 0.01±0.005 | 0.005±0.002 |
| Mixed sediment | 4.03 | 2.32±0.99 | 0.61±0.23 | 36.11±4.04 | 0.005±0.005 | 0.07±0.03 | 0.01±0.005 | 0.02±0.009 |

Rev 2 Comment 2: Lines 154: 12 mm sieve seems like a rather large size when the usual is 2mm. It seems that the authors did not want to disrupt aggregates. That reasoning should be given here.

**Our response:** That is correct. We provided this information in lines 143-144. Initially, we thought that this information would fit better in the sample preparation section than in the experiment design. But we will move this information to section 2.3 and revise lines 154-155 as follows: "Briefly, 50 g of 12 mm sieved air-dried soil were weighed into a 100 ml beaker. Soil samples were sieved to 12 mm to homogenize the substrate while maintaining aggregate structure."

Rev 2 Comment 3: *Line 174: An average of the respiration rates over 120 days, when the rates usually decrease exponentially, seems like an odd metric. Why was this parameter chosen instead of say, cumulative C loss over 120 days?*

**Our response:** We certainly agree with this comment. Cumulative C loss is widely used as a proxy for soil C loss. Please note that we excluded four days as our pre-incubation period to give samples idle time to adjust to rewetting after being stored dry and prepared again for incubation. Thus, we thought that presenting cumulative values might give a biased impression since we were interested in the weighted averages rather than the absolute values of respired C. However, our weighted averages could also be easily converted into cumulative C loss, if the reviewer still considers this to be the better option. For clarification, we will revise lines 175-177 as follows: "As our aim was to compare average respiration between samples rather than the absolute values through the entire period of the experiment. We analyzed data as the weighted average of SPR and TPR over the entire length of the experiment after respiration leveled off. The weight was defined by how many days of the incubation experiment each observation represented."

Rev 2 Comment 4: *Line185: How was the 14C collected from three replicate jars into one evacuated container? Wasn't the vacuum in the container a different strength for each replicate so that they may not have been sampled equally?*

**Our response:** This would indeed yield different strengths for each replicate. Instead of connecting the evacuated container directly to the jars, we collected 120 ml from each replicate using a syringe and transferred the gas in the pre-evacuated container using a tube adapter. This approach is commonly used by the Max Planck Institute for Biogeochemistry in Jena. We will therefore clarify line 185 as follows:

"After accumulation, 120 ml of headspace gas from each field replicate incubation jar was sampled using a syringe. These replicate samples were transferred into a single 400 ml pre-evacuated Restek canister for composite analysis."

Rev 2 Comment 5: *Line 254: How did you evaluate the distinctness of the RCs based on F-values?*

**Our response:** The contributions to SPR and $\Delta^{14}C$ and distinctness of individual rPCs were evaluated based on their p-Values and standardized coefficients. We used the F-statistics to evaluate the explainability of RCs for the three models as reported in Table 2. We will revise line 254 as follows: "We used p-Values (p<0.1) and standardized coefficients to evaluate the explainability of individual RCs while the F-statistic was used to evaluate the overall relationship between RCs and SPR or $\Delta^{14}C$ for every model."

Rev 2 Comment 6: *Section 3.1: To help streamline the manuscript, I recommend getting rid of the discussion of TPR in the results, since SPR is the focus of the manuscript. Perhaps the TPR graphs and language could be in the supplement? I am not sure what additional understanding the TPR variable really adds here.*

**Our response:** While SPR provides important information in C-rich soils, in lower C soils especially subsoils in our study sites, TPR provides additional information that cannot be revealed by SPR. Nevertheless, we agree to revise Figure 1 and keep data related to SPR only. Data related to TPR will be reported in the appendices and only very briefly referred to in the text. Consequently, we will revise the results section so that it is aligned with the revised, more focused figure (See revised figure below).

[Figure]

**Figure 1.** Average and standard errors underline{based on field replicates} for specific potential respiration (SPR) as bars (lower panel) and the C:N ratio as points on top of the bars (upper panel), for non-valley positions (**a**) and valley position (**b**). (N=9 for non-valleys, and N=3 valleys). The same letters on top of bars indicate no significant difference in SPR following ANOVA tested for differences between geochemical regions and depth intervals."x" indicates no significant difference between depth intervals within geochemical regions. ANOVA tests were performed separately for non-valley and valley positions.

Rev 2 Comment 7: *Lines 280-281: I think this sentence is basically a repeat of the first sentence*

**Our response:** This is indeed redundant. In the revised manuscript, we will exclude the following sentence in that paragraph. "For non-valley positions, no statistically significant differences for SPR, TPR, and $\Delta^{14}C$ were found between sloping and plateau positions"

Rev 2 Comment 8: *Fig 1. Are the standard errors based on the replicates or the measurement times since all were averaged to get these values.*

**Our response:** That is correct, the standard errors presented here are based on the field replicates. We will add this information and revise the figure caption as follows: "Figure 1. Average and standard errors based on the field replicates for specific potential respiration (SPR)......."

**Rev 2 Comment 9:** *Fig 2. I think these graphs could better show the differences between the bulk and respired 14C based on how you discuss the results in section 3.2. It would be easier to compare bulk and respired 14C if they were put on the same graph. The way they are now it is hard to see when they are similar and when they are not.*

**Our response:** Thank you for this suggestion, we agree to revise Figure 2 and merge the panels as suggested. See the revised figure and corresponding caption below:

[Figure]

**Figure 2:** Average and standard errors for (a) radiocarbon content ($\Delta^{14}C$) of the bulk soil and respired $CO_2$ for non-valley positions, (b) $\Delta^{14}C$ of the bulk soil and respired $CO_2$ for valley positions, (n=27 for non-valleys, and n=9 valleys for each depth interval).

**Rev 2 Comment 10:** *Lines 345-349: I am not sure what the extrapolation of the respiration rates of the fossil organic C add here and in Table 1. Given the caveats, which you mention in the discussion, it would be better to leave these numbers to the discussion only.*

**Our response:** We agree that highlighting this information in the discussion section would suffice. We will remove lines 345-348 (see the underlined text below) in the results section and revise Table1 as shown below.

"Considering the measured respiration, under the conditions of our lab incubation experiments, we calculate all fossil organic C in non-valley positions would be mineralized in approximately 450 years from topsoil and in 387-440 years from the subsoil. In valley positions, fossil organic C in topsoil would be mineralized after 61 years. In valley subsoil, fossil organic C would be mineralized after 50-100 years."

**Table 1.** Biogenic and fossil organic carbon contribution in the mixed sediment rock region to SOC and respired $CO_2$ as % of total C and ratio bulk soil / respired C for both parameters. Values are displayed separately for non-valley and valley positions per soil depth. (one observation per position due to merging of replicates into composites prior to analysis).

| Position | Depth [cm] | Biogenic [%] Bulk soil | Biogenic [%] Respired gas | Bulk/Respired | Fossil [%] Bulk soil | Fossil [%] Respired gas | Bulk/Respired |
|---|---|---|---|---|---|---|---|
| Non-valley | 0-10 | 89 | 96 | 0.9 | 11 | 4 | 2.8 |
| | 30-40 | 61 | 93 | 0.6 | 39 | 7 | 6.0 |
| | 60-70 | 48 | 91 | 0.5 | 52 | 9 | 5.8 |
| Valley | 0-10 | 98 | 97 | 1.0 | 2 | 3 | 0.7 |
| | 30-40 | 72 | 81 | 0.9 | 28 | 19 | 1.5 |
| | 60-70 | 57 | 61 | 0.9 | 43 | 39 | 1.1 |

Rev 2 Comment 11: *Fig 4. After all the data that is presented in the results, it is odd that the discussion starts off with yet more data! I find figure 4 overwhelming. It has 8 graphs, each with three correlations, with a total of 24 to examine! Many of these are not significant. I suggest saving the whole figure for the supplement and choosing 1-3 graphs to highlight in the discussion. Furthermore, something should indicate which relationships are significant here, maybe make the r and p values bold where they are significant?*

**Our response:** Thank you for this comment. We certainly agree to revise Figure 4. We also agree that having this information in the appendices section is better and could shorten the discussion. We propose the following:

1. We will revise Figure 4 and keep those variables that are largely discussed.
2. We will move Figure 4 into the appendices section and shorten the discussion related to it to the most essential parts (shown in the figure).
3. See the revised figure below.

[Figure]

Figure 4. Pearson correlation between composite of corresponding replicates of $\Delta^{14}C$ of respired $CO_2$ and SPR to P (panels, a-b), and N (panels, c-d) available nutrient data reported by Kidinda et al. (2020 in review) normalized to SOC content for non-valley positions. Bioavailable P = Bray-P, TDN = Total dissolved nitrogen. Data displayed in panels a, and c, are averages plus standard errors of three field replicates. Panels b, and d, show all individual field replicates. Note that two outliers (artifacts) with high bioavailable P values in subsoil were removed from panels a, and b. Pearson correlations and p-values in bold font indicate significant results at p<0.05

Rev 2 Comment 12: *Line 474: I am confused by the attribution of mineral stabilization mechanisms to controlling SPR here as amorphous and crystallized oxides had no relationship to SPR and pyrophosphate-extractable had a positive relationship indicating it was not stabilizing the Carbon.*

**Our response:** We certainly agree. While pedogenic oxides did not show effect on SPR, they control SOC stocks. In fact, a detailed analysis conducted on the same samples (Reichenbach et al., 2021) shows that SOC stocks significantly depend on the amount of pedogenic oxides. "Our data (Fig. A1) suggests that carbon associated with pyrophosphate extractable oxides is readily available to microbial decomposers even in a short-term respiration experiment. In contrast, the presence of oxalate or DCB extractable oxides (and the carbon associated with it) does not relate to the short-term C respiration in our study. Instead, its effects on the long-term SOC stability are more likely related to the formation of stable aggregates (Oades, 1988, Kleber et al., 2005, Reichenbach et al. 2021). This is the main message we wanted to share in Fig. A1."

In the revised manuscript we proposed to add the above statement between lines (464 and 465) in order to clarify this relationship.

**References added:**

Oades, J. M.: The retention of organic matter in soils, Biogeochemistry, 5, 35-70, https://doi.org/10.1007/BF02180317,1988.

Martinez, P., and Souza, I. F.: Genesis of pseudo-sand structure in Oxisols from Brazil – a review, Geoderma Regional [pre-proof], https://doi.org/10.1016/j.geodrs.2020.e00292, 2019.

Reichenbach, M., Fiener, P., Garland, G., Griepentrog, M., Six, J. and Doetterl, S.: The role of geochemistry in organic carbon stabilization in tropical rainforest soils, SOIL [in review], 1–35, doi:10.5194/soil-2020-92, 2021.

Kirsten, M., Mikutta, R., Vogel, C., Thompson, A., Mueller, C. W., Kimaro, D. N., Bergsma, H. L. T., Feger, K. H. and Kalbitz, K.: Iron oxides and aluminous clays selectively control soil carbon storage and stability in the humid tropics, Sci. Rep., 11(1), 1–12, doi:10.1038/s41598-021-84777-7, 2021.

*Rev 2 Comment 13: Fig 5. Can you bold the p values for what is significant here? Same for the similar graphs in the Appendix.*

**Our response:** Thank you for this suggestion. We will revise Figure 5, and Figure A2 and put in bold significant correlations and their corresponding p-values, and amend the figure captions accordingly.

Rev 2 Comment 14: *502: specify high C:N here*

**Our response:** We will revise this paragraph as follow:

"Despite similar SOC stocks, SPR and TPR were lowest in soils of the mixed sediment region, which also had the lowest bulk soil and respired $\Delta^{14}$C of the three geochemical regions (Fig. 1, 2). The depletion of N and high C:N values (153.9 ± 68.5) of fossil organic C likely contributed to reducing soil respiration rates in the mixed sediment region (Whitaker et al., 2014)."

**Reference added:**

Whitaker, J., Ostle, N., McNamara, N. P., Nottingham, A. T., Stott, A. W., Bardgett, R. D., Salinas, N., Ccahuana, A. J. Q. and Meir, P.: Microbial carbon mineralization in tropical lowland and montane forest soils of Peru, Front. Microbiol. , 5, 720: https://www.frontiersin.org/article/10.3389/fmicb.2014.00720, 2014.

---

## Author Response (AR1)

**Response letter to the reviewers' comments for the SOIL manuscript**

Dear SOIL Editors, Dear Reviewers,

We would like to thank you for the time to evaluate our manuscript (entitled "**Heterotrophic soil respiration and carbon cycling in geochemically distinct African tropical forest soils**" with reference number "Soil-2020-96". We are very pleased that the reviewers assessed our work positively and recognized its potential and, especially, its significant contribution to the soil organic carbon research in tropical Africa. The comments and suggestions provided by the reviewers and the topical editor helped greatly to improve our manuscript and we would like to thank you all for the constructive and valuable insights. We have addressed all comments and suggestions you provided to the best of our ability. In particular, the two reviewers offered suggestions on how to shorten our results and discussion sections and how to streamline our manuscript. Briefly, we have addressed all comments and responded to all suggestions as follow:

- We restructured and extended the introduction section, removed subheaders
and added new information about the role of nutrients in C cycling

- We revised our results section and the accompanying figures and kept only those figures and tables that are necessary to understand the manuscript.

- We have shortened and restructured the discussion section to streamline the manuscript towards the main research question and hypothesis and tried to put our results more into a larger context.

Please find below a point-by-point response to all the reviewers' comments and how we addressed them. We have submitted the revised document and also a track-change version to facilitate the review process of the implemented changes. Reviewer comments are in italic, our responses are listed always directly afterwards. Suggested text that we have added or removed in the revised manuscript is stated between " " and new/changed text underlined. Lines provided by the reviewers refer to the earlier version of the manuscript while lines and references given in our responses refer to the revised finalized manuscript.

We hope you find our responses and changes to the manuscript satisfying and we are looking forward to hearing your opinion on our revised manuscript.

Yours sincerely,

The authors

**Reviewer 1**

**Point-by-Point response**

*The article "Controls on heterotrophic soil respiration and carbon cycling in geochemically distinct African tropical forests soils" investigates the role of soil chemistry, fertility and geochemical composition as drivers of soil respiration under laboratory conditions in soils collected along slope gradients in tropical Africa. The article fits the scope of the journal and it will be of great interest for the journal readers.*

**Our response:** We thank the reviewer for the overall positive evaluation, comments, and suggestions.

Rev 1 Comment 1: *The introduction is to large extents well structured, though I am not sure if the subheaders are really needed.*

**Our response:** Thank you for this recommendation. We have removed the two subheaders and connected the two sections with a new paragraph introducing the role of nutrients on C cycling (lines 66-75).

Rev 1 Comment 2: *The introduction contains a lot of information on geochemical (e.g. Al Fe SiO2) parameters influencing soil C dynamics, but not so much on available P or N, which turn out in this study to be strong determinants of soil respiration.*

**Our response:** We certainly agree. While the role of nutrients on carbon input and stock was briefly introduced in the early version of the manuscript, the role of nutrients and microbial activity on soil carbon respiration could be further discussed. We made the following revisions: We extended the introduction and added information on nutrients in the revised manuscript, especially between lines 66-75 (see the suggested text underlined below). We added information related to the work of (Fernández-Martínez et al., 2014; Liu et al., 2015; Jing et al., 2020; Kirsten et al., 2021; and Kallenbach et al., 2016, Mikutta et al., 2019).

"Furthermore, long-term chemical weathering in tropical systems has led to the depletion of rock-derived nutrients in soils and has limited the capacity of microorganisms and plants to access these nutrients (Liu et al., 2015; Vitousek and Chadwick, 2013). It is likely that variation in soil weathering stage and nutrient availability in tropical forests affect soil C storage and the exchange of C between plants, soil and the atmosphere. For example, due to their tight coupling driven by the metabolic needs of plants and microorganisms, changes in nutrient availability such as nitrogen (N) and phosphorus (P) can greatly alter the terrestrial C cycle, partly because $CO_2$ uptake by terrestrial ecosystems strongly depends on N and P availability (Fernández-Martínez et al., 2014). Furthermore, low N and P availability limits microbial growth and activities and therefore affects the cycling of organic matter (Jing et al., 2020; Liu et al., 2015)."

**References added:**

Fernández-Martínez, M., Vicca, S., Janssens, I. A., Sardans, J., Luyssaert, S., Campioli, M., Chapin Iii, F. S., Ciais, P., Malhi, Y., Obersteiner, M., Papale, D., Piao, S. L., Reichstein, M., Rodà, F. and Peñuelas, J.: Nutrient

availability as the key regulator of global forest carbon balance, Nat. Clim. Chang. |, 4, doi:10.1038/NCLIMATE2177, 2014.

Jing, X., Chen, X., Fang, J., Ji, C., Shen, H., Zheng, C. and Zhu, B.: Soil microbial carbon and nutrient constraints are driven more by climate and soil physicochemical properties than by nutrient addition in forest ecosystems, Soil Biol. Biochem., 141, 107657, doi:10.1016/j.soilbio.2019.107657, 2020.

Liu, L., Gundersen, P., Zhang, W., Zhang, T., Chen, H. and Mo, J.: Effects of nitrogen and phosphorus additions on soil microbial biomass and community structure in two reforested tropical forests, Sci. Rep., 5, doi:10.1038/srep14378, 2015

Rev 1 Comment 3: *Moreover, little information is given in the introduction on the role of aggregation, or microbial biomass as C sequestration 'pump' (or also about microbial enzymatic control). I would recommend extending each of the topics a bit more to make the introduction to make the link even stronger towards the research questions and to the results presented.*

**Our response:** We agree with the reviewer that this could be further strengthened. We have added information in relation to the work of Fang et al. (2017), Rasmussen et al. (2018), Kirsten et al. (2021), von Fromm et al. (2021), and revised this section in lines 80-85 as follows:

"In contrast, stable microaggregates rich in iron (Fe) and aluminum (Al) oxyhydroxides found in abundance in tropical soils (Bruun et al., 2010; Torres-Sallan et al., 2017) seem to be of greater importance in stabilizing C in tropical soils, as concentrations of Al and Fe are commonly higher than in many temperate soils (Khomo et al., 2017). This is confirmed by studies conducted across a wide range of tropical ecoregions showing that SOC is mainly regulated by Fe or Al (hydr) oxides, more so than by clay content (Fang et al., 2019; von Fromm et al., 2021; Rasmussen et al., 2018)".

**References added:**

Torres-Sallan, G., Schulte, R. P. O., Lanigan, G. J., Byrne, K. A., Reidy, B., Simó, I., Six, J. and Creamer, R. E.: Clay illuviation provides a long-term sink for C sequestration in subsoils, Sci. Rep., 7(1), 45635, doi:10.1038/srep45635, 2017.

Rasmussen, C., Heckman, K., Wieder, W. R., Keiluweit, M., Lawrence, C. R., Berhe, A. A., Blankinship, J. C., Crow, S. E., Druhan, J. L., Hicks Pries, C. E., Marin-Spiotta, E., Plante, A. F., Schädel, C., Schimel, J. P., Sierra, C. A., Thompson, A. and Wagai, R.: Beyond clay: towards an improved set of variables for predicting soil organic matter content, Biogeochemistry, 137, 297–306, doi:10.1007/s10533-018-0424-3, 2018.

Fang, K., Qin, S., Chen, L., Zhang, Q. and Yang, Y.: Al/Fe Mineral Controls on Soil Organic Carbon Stock Across Tibetan Alpine Grasslands, J. Geophys. Res. Biogeosciences, 124, 247–259, doi:10.1029/2018JG004782, 2019.

Khomo, L., Trumbore, S. E., Bern, C. R. and Chadwick, O. A.: Timescales of carbon turnover in soils with mixed crystalline mineralogies, SOIL, 3, 17–30, doi:10.5194/soil-3-17-2017, 2017.

Rev1 Comment 4: *In addition, I think the hypotheses could be more specifically state, e.g. in line 93 to 96 it would be helpful to mention which change in geochemical properties would cause which response by microbial decomposers more 'explicitly', by stating the expected mechanism, or by hypothesizing under which conditions faster or slower soil C turnover could happen, and how this has been influencing soil C stocks in the long term.*

**Our response:** We have revised the hypotheses following the key points suggested. We revised lines 93-100 as follows: "hypothesize that (1) specific soil respiration and the $\Delta^{14}C$ signature of potential soil respiration in tropical soils are primarily controlled by geochemical properties related to soil fertility derived from and varying with soil parent material. These variations in soil fertility can stimulate or inhibit microbial activity and increase or decrease soil C decomposition rates. (2) The presence or absence of C stabilization mechanisms, in soils, related to mineral geochemistry and soil formation, can increase SOC stocks and decrease heterotrophic C respiration rates by creating an energetic barrier for C decomposers, for example through complexation with organic molecules or by forming stable (micro) aggregates. (3) The topographic origin of a soil sample controls specific soil respiration and its $\Delta^{14}C$ signature indirectly through the environmental conditions under which soil C decomposition took place in situ, modifying the quality and quantity of the available SOC stock prior to the experiment."

Rev 1 Comment 5: *The material and methods provide a detailed characterization of the study sites and the respective soil properties including many references to articles that are currently in review, which is a bit difficult to trace. The incubation experiment setup is very clearly described and sound. Also, the statistical analysis is provided in detail, which is great. One minor point that I could suggest to improve the role of soil depth would be to explore linear models and include soil depth 'nested' into topographic position and geochemical region (or nested per sample location), as the different depths are not independent of each other.*

**Our response:** Thank you for this suggestion. We have revised the description of the study sites and presented the chemical characteristics in the form of a table as suggested by the second reviewer as well. Regarding the statistics and the nested approach, we explored this approach but it did not improve our results. We tested two generalized linear models: one with all factors including geochemical regions, depth intervals, and their interaction; and another one where depth intervals are nested within geochemical regions. We compared the two models using a one-way ANOVA tests. The result presented in the table below suggest that nesting did not improve the results.

Table: Analysis of deviance between Model 1 (where depth intervals are nested within geochemical region) and Model 2 (where we consider interaction of geochemical region and depth intervals). The One-way ANOVA() function performs a test comparing the two models. Here the associated deviance is nearly zero. This suggests that model 2 containing the interaction is indifferent in its performance to model 1 with depth interval nested within geochemical region.

| Analysis of Deviance | | | | |
|---|---|---|---|---|
| Model 1: sqrt(SPR) ~ region/Depth Interval | | | | |
| Model 2: sqrt(SPR) ~ region: Depth Interval | | | | |
| | Resid. Df | Resid. Dev | Df | Deviance |
| 1 | 76 | 6.6006 | | |
| 2 | 76 | 6.6006 | 0 | 1.7764e-15 |

Rev 1 Comment 6: *The results are well described, some specific suggestions are given below, the figures and tables are adequate. The discussion section is relative to the other parts quite long and reads a bit lengthy. It could maybe be a bit shorter or more streamlined towards the initial research questions and hypotheses. For me, surprising was that in the discussion section further analyses were presented that appear rather as an extension of the results, and may in my opinion therefore also maybe be rather moved to the results section.*

**Our response:** Thank you for this comment, we have revised, and shortened the discussion section in the revised manuscript. Since the discussion has been greatly restructured, we will not paste all changes here. In this response here we only highlight important changes made in this section. For details we refer the reviewers to lines 408-485 of the discussion section.

- Discussion has been streamlined towards answering the initial research questions in the light of our results and comparative literature. No new data is introduced anymore in the discussion.
- To shorten our discussion, we have removed all figures in the discussion section. We moved all figures from the discussion to the appendices and changed their captions accordingly
- We revised figure 4 and 5 and renamed it to Figure A1 and A2 and kept only those parameter that are actually discussed in the paper
- We created new subheaders, and put our results in larger context in relation to other studies conducted in comparable tropical systems

Rev 1 Comment 7: *In addition, it includes the analysis of a much larger set of parameters, which have not really been introduced before (e.g. dissolved organic C, bioavailable P, or enzymes), and I thought, which would already be included in the rPCA analysis? Maybe it would be better to also move the graphs in the appendix section, as they rather support already stated emerging patterns.*

**Our response:** Thank you for this comment. We agree that having this information in the appendices would shorten the discussion section and improve the readability. In the revised manuscript, we have moved Figure 4 into the appendix section, renamed it to Figure A1 and revised it to the most essential parts related to the discussion. We believe that these changes will create a direct connection between discussion sections and improve readability. Figure A1 presents variables initially included in the rPCA and represented by the RCs. While the RCs address issues like autocorrelations and model overfitting, they don't provide information on direct relationships between initial variables. In Figure A1 we try to present these relationships masked by RCs.

Rev 1 Comment 8: *In addition, terminology changes also a bit (e.g. mixed region vs. sediment region) – double check, please.*

**Our response:** Thank you for noting this. Based on the elemental composition of the bedrock (see methods), we classified our study area into three geochemical regions including the mafic region, felsic region, and mixed sediment region. We have addressed any inconsistencies related to the names of the geochemical regions throughout the manuscript.

Rev 1 Comment 9: *Moreover, large parts of the discussion are rather discussion the results of the study (see in the technical comments), but there could be more discussion relating the findings to other results found in other tropical/subtropical/montane forest sites and put the results more in to a larger context. Finally, in the discussion a large part is about microbial nutrient limitation, it would be great to already introduce this as a possible control in the introduction in more detail.*

**Our response:** Since this comment is connected to comment 6, please see the comment and response above. Here, we highlight the changes we made in the discussion. For details, we would like to refer the reviewer to lines 408-485 of the discussion section and lines 66-76 of the introduction section.

- We have introduced the concept of potential nutrient limitations in tropical soils in lines 66-76.
- We have created new subheaders, and revised the discussion section the most import information related to the research question and hypotheses
- We have streamlined our results and corresponding discussion in larger context in relation to other studies conducted in tropical systems

Rev 1 Comment 10: *Line 77: this could need a reference.*

**Our response:** We have added reference in line 75, to support this statement following information from Addo-Danso et al. (2018).

**Reference added:**

Addo-Danso, S. D., Prescott, C. E., Adu-Bredu, S., Duah-Gyamfi, A., Moore, S., Guy, R. D., Forrester, D. I., Owusu-Afriyie, K., Marshall, P. L. and Malhi, Y.: Fine-root exploitation strategies differ in tropical old growth and logged-over forests in Ghana, Biotropica, 50(4), 606–615, doi:10.1111/btp.12556, 2018.

Rev 1 Comment 11: *Line 222: could you provide a reference for subsoil conditions?*

**Our response:** Thank you for this comment. This line has been excluded in the revised document.

Rev 1 Comment 12: *Line 255: I guess there were no real differences between the plateau, mid-slope, slope positions – still it would be great to mention why these different locations were not considered anymore.*

**Our response:** That is correct, there were no differences between plateau, upper slope, and midslope. As this was a result of statistical analysis, in the early version of the manuscript we presented this information in the first section of our results. Since this could cause confusion about our statistical approach in the methods section, in the revised manuscript, we decided to move this paragraph back to the statistical analysis section in methods. See lines 260-264.

"Note that we found no statistical difference in SPR or $\Delta^{14}C$ between the plateau and slope positions within each studied geochemical region (mafic, felsic, and mixed sediment). Across geochemical regions and soil depths, SPR, and $\Delta^{14}C$ differed only between valleys and non-valley positions. Hence, all further analyses were done after splitting the data into two subsets: (1) non-valley positions (plateau, upper slope, and middle slope) versus (2) valley positions (valleys and foot slope)."

Rev 1 Comment 13: *Line 281: The sentence 'Within non-valley positions…' is redundant.*

**Our response:** Thank you for noting this. We have addressed this in the revised version

Rev 1 Comment 14: *Line 299-301: Does this describe exactly the same as is stated in Line 286-288?*

**Our response:** Our aim was to describe the results at the levels of topography and geochemical regions separately. However, the two paragraphs provide indeed the same information. To avoid repetition in the revised version, we excluded the line below and described the results following the updated figures:
"Differences in the subsoil, however, were not significant across regions (Fig. 1C). $\Delta^{14}C$ activity of both soil and respired $CO_2$ in the mafic and felsic regions were not significantly different from each other for both top- and subsoil."

Rev 1 Comment 15: *Line 315: I don't understand the x indicating no significant difference between depth intervals within geochemical regions, I am not sure which differences the letters demonstrate – within regions or across all regions and depths. Please can you clarify this?*

**Our response:** For the ANOVA analysis, we did pairwise comparisons between soil depth intervals within geochemical regions and between soil depth intervals across geochemical regions. For Fig.1, x indicates "no significant difference between depth intervals within geochemical region. ANOVA tests were performed separately for non-valley and valley positions."

Rev 1 Comment 16: *Line 324: should there not be two different results? Or should this indicate the Δ14C of bulk soil and of respired CO2 were highly correlated.*

**Our response:** The sentence presents one result "the relationship between $\Delta^{14}C$ of bulk soil and $\Delta^{14}C$ respired $CO_2$". However, we agree that it might be confusing. We have revised this line (329-330) as follows: Across all study regions we found a strong relationship ($R^2 = 0.81$, p<0.1) between $\Delta^{14}C$ of the bulk soil and $\Delta^{14}C$ of the respired $CO_2$.

Rev 1 Comment 17: *Line 393: delete 'from it'.*

**Our response:** Thank you for noting this, we have corrected this in the revised version.

Rev 1 Comment 18: *Line 429. Maybe introduce indicators for N & P limitation of microbial decomposers a bit earlier already.*

**Our response:** We have added information in the introduction section, highlighting N and P limitations to microbial decomposers. In the revised document, we have added the statement below line 69-79:

"For example, due to their tight coupling driven by the metabolic needs of plants and microorganisms, changes in nutrient availability such as nitrogen (N) and phosphorus (P) can greatly alter the terrestrial C cycle, partly because CO2 uptake by terrestrial ecosystems strongly depends on N and P availability (Fernández-Martínez et al., 2014). Furthermore, low N and P availability limits microbial growth and activities and therefore affects the cycling of organic matter (Jing et al., 2020; Liu et al., 2015).

Rev 1 Comment 19: *Line 429: Could you repeat what is considered as poor quality? (e.g. CN ratios of soil organic matter or any other parameters?)*

**Our response:** We used lower nitrogen and higher C:N values to characterize the poor quality of SOC in the mixed sediment region. To clarify this statement, we revised this statement in lines 419-421 as follows: "In addition, the depletion of N and high C:N values (153.9 ± 68.5) of fossil organic C, which encompasses a

substantial part of total C in subsoils of the mixed sediment region (Table 2), was likely an additional factor reducing soil respiration rates (Whitaker et al., 2014)."

**Reference added:**

Whitaker, J., Ostle, N., McNamara, N. P., Nottingham, A. T., Stott, A. W., Bardgett, R. D., Salinas, N., Ccahuana, A. J. Q. and Meir, P.: Microbial carbon mineralization in tropical lowland and montane forest soils of Peru, Front. Microbiol. , 5, 720: https://www.frontiersin.org/article/10.3389/fmicb.2014.00720, 2014.

Rev 1 Comment 20: *Line 434: check sentence – lower compared to what – and check tenses – fossil C content 'was' low.*

**Our response:** We revised this statement in line 420-421 as follows: "However, respiration rates in the topsoil of the mixed sediment region were lower compared to the mafic or felsic region (Fig. 1), while fossil organic C content in the topsoil was low compared to the subsoil (Table 2)."

Rev 1 Comment 21: *Line 439: is there maybe also another study that shows that organo-mineral complexation could be saturated depending on which organo mineral complexes are present in soil (e.g. Quesada 2020, Dötterl 2018).*

**Our response:** After revising and restructuring the discussion, this line does not appear in revised manuscript

Rev 1 Comment 22: *Line 443: add after the brackets: ), in our study aggregation…*

**Our response:** Thank you for noting this, we have corrected this sentence.

Rev 1 Comment 23: *Line 460: I would recommend to put this entire section (The role of mineral related C stabilization mechanisms) more into relation with other studies, at the moment, it is rather focusing on either studies from the same data set and reads a bit as an extended results section and could be shortened a bit.*

**Our response:** We have revised and shortened this section (see lines 426-458) and put it in a general context in relation to other studies conducted in tropical systems. We have removed the two subheaders "The role of tropical weathering in explaining soil respiration" and "The role of mineral related C stabilization mechanisms". We replaced them with one new subheader "The role of tropical weathering and mineral related C stabilization mechanisms in explaining soil respiration".

Rev 1 Comment 24: *Line 506 the same suggestion as above, I think this section can be shortened too, screen for repeated results.*

**Our response:** We revised lines 460:468 as follows:

"The presence of fossil organic C in the mixed sediment region (up to 52% of SOC stock in deeper subsoil) (Table 2), had a marked effect on SOC stocks in subsoils that would otherwise be similarly low to those of the felsic region (Fig. 1c). Consistent with this finding, a recent study shows that fossil organic C can largely contribute to SOC in subsoils (Kalks et al., 2020). While fossil organic C in our study region is of poor quality as indicated by depleted N and high C:N values (153.9 ± 68.5), our data shows that fossil organic C was still microbially available (Fig. 2), leading to the respiration of $CO_2$ with comparably old $^{14}C$ signatures. However, we were unable to quantitatively disentangle the slower biogenic C cycling from the contribution of fossil organic C using $^{14}CO_2$. Thus, whether the presence of FOC and/or other unfavorable chemical soil characteristics in the mixed sediment region contributed to a general slowing of C cycling remains unknown."

Rev 1 Comment 25: *Line 538: check tenses – there is sometimes a switch between present and past tense within sentences.*

**Our response:** Thank you for noting this. "Observe" should be "observed". We have corrected this sentence (line 489) and checked for similar inconsistencies throughout the manuscript.

Rev 1 Comment 26: *Line 570: namely twice in the same paragraph*

**Our response:** We revised the whole section and corrected this statement as follows : "Our results, linked to those of Reichenbach et al. (2021), show that the presence or absence of mineral stabilization mechanisms is particularly important for long-term soil C stocks in tropical soils, varying largely with soil parent material, while short-term respiration relies on readily available C sources".

**Reviewer 2**

**Point-by-Point response**

*In this manuscript, the authors present the results of a 120 day soil incubation study of three soil types (mafic, felsic, and mixed sediment) sampled across topographic positions and investigate the biological and chemical controls on respiration rates and respired 14CO2. This manuscript is a significant contribution to the soil organic carbon research because the soils are from tropical Africa, a region where few studies have been carried out. Furthermore, this manuscript looks at the important C stabilization mechanisms in highly weathered soils, which are also poorly sampled and understood relative to less weathered soils often found in temperate regions. Another positive is that the researchers examined 3 depths within the soil profile. I found no glaring issues with the methods used in this study. I did have some questions about the way some of the data were presented. On one hand, this paper could be shorter and more streamlined, but on the other hand the lack of data from Tropical Africa does greatly increase the value of all the parameters reported here. I offer some suggestions on how to shorten the paper below.*

**Our response:** We thank the reviewer for the overall positive evaluation, comments, and suggestions.

Rev 2 Comment 1: *Lines 107-125: This whole paragraph would be better as a Table introducing the sites and their location and chemistry. A table would be easier to read than a paragraph and make it easier to compare the sites.*

**Our response:** We agree that this section could be shortened by adding a summary table. We have summarized the chemical composition of the sites in the form of a table (see table below) and revised lines 112-125 as follows:

"Study sites in the DRC are located in Kahuzi-Biega National Park (-2.31439° S; 28.75246° E) where soils have developed from mafic magmatic rocks, a result of volcanism in the East African Rift System (Schlüter, 2006). Mafic magmatic rocks in the region are characterized by high Fe and Al and low Si content as well as a high content of rock-derived nutrients such as base cations, and P (Table 1). Study sites in Uganda are located in Kibale National Park (0.46225° N; 30.37403° E) where soils have developed from felsic magmatic and metamorphic rocks. The felsic magmatic rocks in our study region are characterized by the gneissic-granulitic complex with low contents of Fe, and Al, and high Si content. Unlike mafic, felsic magmatic rocks in our study sites are characterized by low content of rock-derived nutrients (Table 1). Study sites in Rwanda are located in Nyungwe National Park (-2.463088° S; 29.103834° E) where soils have developed from a mixture of sedimentary rocks of varying geochemistry. These sediments are mostly dominated by quartz-rich sandstones and schist layers spanning along the Congo-Nile divide in the western province of Rwanda (Schlüter, 2006). Similar to the felsic magmatic soils, mixed sediments in our study sites are characterized by low Fe and Al content but high Si content and low content of rock-derived nutrients. A specific feature of the mixed sedimentary rocks in our study region

is the presence of fossil organic carbon (Table 1). Fossil organic carbon in these sediments is further characterized by a high C:N ratio (153.9 ± 68.5), and is depleted in N (Doetterl et al., 2021; Reichenbach et al., 2021)"

**Table 1.** Chemical composition of unweathered rock samples representing the soil parent material in the investigated three geochemical regions. Values represent mean ± standard errors (N=6, 10 and 3 for mafic, felsic and mixed sediment respectively).

| Geochemical region | C[%] | Fe [%] | Al [%] | Si [%] | Ca [%] | K [%] | Mg [%] | P [%] |
|---|---|---|---|---|---|---|---|---|
| Mafic | 0 | 8.98 ±0.75 | 6.26±1.15 | 14.22±0.82 | 0.58±0.23 | 0.08±0.03 | 1.25±0.13 | 0.36±0.05 |
| Felic | 0 | 1.08±0.5 | 0.51±0.38 | 37.28±1.87 | 0.01±0.004 | 0.01±0.006 | 0.01±0.005 | 0.005±0.002 |
| Mixed sediment | 4.03 | 2.32±0.99 | 0.61±0.23 | 36.11±4.04 | 0.005±0.005 | 0.07±0.03 | 0.01±0.005 | 0.02±0.009 |

Rev 2 Comment 2: Lines 154: 12 mm sieve seems like a rather large size when the usual is 2mm. It seems that the authors did not want to disrupt aggregates. That reasoning should be given here.

**Our response:** That is correct. But we added this information in section 2.3 and revise lines 161-1163 as follows: "Briefly, 50 g of 12 mm sieved air-dried soil were weighed into a 100 ml beaker. Soil samples were sieved to 12 mm to homogenize the substrate while maintaining aggregate structure at a low level of disturbance."

Rev 2 Comment 3: *Line 174: An average of the respiration rates over 120 days, when the rates usually decrease exponentially, seems like an odd metric. Why was this parameter chosen instead of say, cumulative C loss over 120 days?*

**Our response:** We certainly agree with this comment. Cumulative C loss is widely used as a proxy for soil C loss. Please note that we excluded four days as our pre-incubation period to give samples idle time to adjust to rewetting after being stored dry and prepared again for incubation. Thus, we thought that presenting cumulative values might give a biased impression since we were interested in the weighted averages rather than the absolute values of respired C. However, our weighted averages could also be easily converted into cumulative C loss, if the reviewer still considers this to be the better option. For clarification, we have revised lines 184-186 as follows: "Our aim was to compare average respiration between samples rather than the absolute values through the entire period of the experiment. Thus, we analysed data as the weighted average of SPR over the entire length of the experiment after respiration levelled off. We defined the weight by how many days of the incubation experiment each observation represents."

Rev 2 Comment 4: *Line185: How was the 14C collected from three replicate jars into one evacuated container? Wasn't the vacuum in the container a different strength for each replicate so that they may not have been sampled equally?*

**Our response:** This would indeed yield different strengths for each replicate. Instead of connecting the evacuated container directly to the jars, we collected 120 ml from each replicate using a syringe and transferred the gas in the pre-evacuated container using a tube adapter. This approach is commonly used by the Max Planck Institute for Biogeochemistry in Jena. We have clarified this and revised line 201-202 as follows:

"After accumulation, 120 ml of headspace gas from each field replicate incubation jar was sampled using a syringe. These replicate samples were transferred into a single 400 ml pre-evacuated Restek canister for composite analysis."

Rev 2 Comment 5: *Line 254: How did you evaluate the distinctness of the RCs based on F-values?*

**Our response:** The contributions to SPR and $\Delta^{14}C$ and distinctness of individual rPCs were evaluated based on their p-Values and standardized coefficients. We used the F-statistics to evaluate the explainability of RCs for the three models as reported in Table 3. We have revised line 258-259 as follows: "We used p-Values (p<0.1) and standardized coefficients to evaluate the contribution of explanatory power of individual RCs to the overall model while the F-statistic was used to evaluate the overall relationship between RCs and SPR or $\Delta^{14}C$ for every model."

Rev 2 Comment 6: *Section 3.1: To help streamline the manuscript, I recommend getting rid of the discussion of TPR in the results, since SPR is the focus of the manuscript. Perhaps the TPR graphs and language could be in the supplement? I am not sure what additional understanding the TPR variable really adds here.*

**Our response:** While SPR provides important information in C-rich soils, in soils with lower C content, especially subsoils in our study sites, TPR provides additional information that cannot be revealed by SPR. Nevertheless, we have revised Figure 1 and kept data related to revised discussion. Data related to TPR were removed from this manuscript but could be still calculated from interested readers by linking the presented SPR and SOC data. Consequently, we have revised the results section so that it is aligned with the revised discussion, more focused figure (See revised figure below).

[Figure]

**Figure 1.** Average and standard errors based on field replicates, (**a**) C:N ratio as points (top) and specific potential respiration (SPR) bottom for non-valley positions (N = 9), (**b**) C:N ratio as points (top) and specific potential respiration (SPR) bottom for valley positions (n= 3), (**c**) SOC stocks and (**d**) Bioavailable phosphorus for non-valley positions (n = 9). Same letters on top of bars indicate no significant difference following ANOVA tested for differences between geochemical regions and depth intervals. "x" indicates no significant difference between depth intervals within geochemical regions. ANOVA tests were performed separately for non-valley and valley positions.

Rev 2 Comment 7: *Lines 280-281: I think this sentence is basically a repeat of the first sentence*

**Our response:** This is indeed a redundancy. In the revised manuscript, we excluded the following sentence in that paragraph.

Rev 2 Comment 8: *Fig 1. Are the standard errors based on the replicates or the measurement times since all were averaged to get these values.*

**Our response:** That is correct, the standard errors presented here are based on the field replicates. We have added this information and revised the figure caption as follows: "Figure 1. Average and standard errors based on the field replicates for specific potential respiration (SPR)......."

Rev 2 Comment 9: *Fig 2. I think these graphs could better show the differences between the bulk and respired 14C based on how you discuss the results in section 3.2. It would be easier to compare bulk and respired 14C if they were put on the same graph. The way they are now it is hard to see when they are similar and when they are not.*

**Our response:** Thank you for this suggestion, we have revised Figure 2 and merged the panels as suggested. See the revised figure and corresponding caption below:

[Figure]

**Figure 2.** Average and standard errors based on all composite samples for non-valley positions only. (**a**) radiocarbon content ($\Delta^{14}C$) of the bulk soil and respired $CO_2$ for non-valley positions, (**b**) $\Delta^{14}C$ of the bulk soil and respired $CO_2$ for valley positions, (n = 27 for non-valleys, and n = 9 valleys for each depth interval). Note that at non-valley positions, each point in panel 2a represents 3 observations from composite samples. At valley positions, each point in panel 2b represent 1 observation from composite samples.

Rev 2 Comment 10: *Lines 345-349: I am not sure what the extrapolation of the respiration rates of the fossil organic C add here and in Table 1. Given the caveats, which you mention in the discussion, it would be better to leave these numbers to the discussion only.*

**Our response:** We agree that highlighting this information in the discussion section would suffice. We have removed that information in the results section and revised Table2 as shown below.

**Table 2.** Biogenic and fossil organic carbon contribution in the mixed sediment region to SOC and respired $CO_2$ as % of total C and ratio bulk soil / respired C for both parameters. Values are displayed separately for non-valley and valley positions per soil depth (n = 1 per soil depth and position due to merging of replicates into composites prior to analysis). We note that these values are an upper bound on the contribution of fossil organic C, as these estimates may be affected by variable rates of biogenic C cycling.

| Position | Depth [cm] | Biogenic [%] | | | Fossil [%] | | |
|---|---|---|---|---|---|---|---|
| | | Bulk soil | Respired gas | Bulk/Respired | Bulk soil | Respired gas | Bulk/Respired |
| Non-valley | 0-10 | 89 | 96 | 0.9 | 11 | 4 | 2.8 |
| | 30-40 | 61 | 93 | 0.6 | 39 | 7 | 6.0 |
| | 60-70 | 48 | 91 | 0.5 | 52 | 9 | 5.8 |
| Valley | 0-10 | 98 | 97 | 1.0 | 2 | 3 | 0.7 |
| | 30-40 | 72 | 81 | 0.9 | 28 | 19 | 1.5 |
| | 60-70 | 57 | 61 | 0.9 | 43 | 39 | 1.1 |

Rev 2 Comment 11: *Fig 4. After all the data that is presented in the results, it is odd that the discussion starts off with yet more data! I find figure 4 overwhelming. It has 8 graphs, each with three correlations, with a total of 24 to examine! Many of these are not significant. I suggest saving the whole figure for the supplement and choosing 1-3 graphs to highlight in the discussion. Furthermore, something should indicate which relationships are significant here, maybe make the r and p values bold where they are significant?*

**Our response:** We certainly agree, to shorten the discussion, we have moved this information in the appendices section and only present the most necessary data from our parallel study (Kidinda et al. 2020, introduced as such in the methods section) that can support our discussion in a meaningful way. Thus, we have greatly revised Figure 4 (now Figure A1) and kept those variables that are discussed in the manuscript.

[Figure]

**Figure A1.** Pearson correlation between composite of corresponding replicates of $\Delta^{14}C$ of respired $CO_2$ and SPR to P (panels, a-b), and N (panels, c-d) available nutrient data reported by Kidinda et al., (2020) normalized to SOC content for non-valley positions. Data displayed in panels a, and c, are averages plus standard errors of three field replicates. Panels b, and d, show all individual field replicates. Note that two outliers (artefacts) with high bioavailable P values in subsoil were removed from panels a, and b. p-values in bold font indicate significant results at p<0.05. Abbreviations: Bioavailable P = Bray-P, TDN = Total dissolved nitrogen.

Rev 2 Comment 12: *Line 474: I am confused by the attribution of mineral stabilization mechanisms to controlling SPR here as amorphous and crystallized oxides had no relationship to SPR and pyrophosphate-extractable had a positive relationship indicating it was not stabilizing the Carbon.*

**Our response:** We certainly agree that this statement needs clarification. While oxalate extractable pedogenic oxides did not show effect on SPR in our short-term incubation experiment, they can control SOC stocks in long-term. In fact, a detailed analysis conducted on the same samples (Reichenbach et al., 2021) shows that SOC stocks significantly depend on the amount of these oxides. Pyrophosphate extractable oxides represent a different type of bonding between metals and organic matter that shows a positive relationship to SPR, indicating that C bond in soil this way is still readily available. In the revised manuscript, we have clarified this statement in lines 439-

456 as follows: "Comparing our findings on SPR to the abundance of oxalate or DCB extractable Fe or Al amorphous and crystalline pedogenic oxides reported by Reichenbach et al. (2021), we found weak to no correlation (Fig. A3 b-c). We interpret this result as an indication that C stabilized by such minerals does not contribute to soil respiration in a significant way in our short-term respiration experiment. Its effects on the long-term SOC stability are more likely related to the formation of stable aggregates (Kleber et al., 2005; Oades, 1988;Barthès et al., 2008; Rasmussen et al., 2018; Traoré et al., 2020; Quesada et al., 2020). Stable metal-organic complexes then represent energetic barriers in soil that are hard to overcome for microorganisms to access potential C resources (Bruun et al., 2010; Zech et al., 1997). The importance of these mechanisms is illustrated by the fact that although mafic soils were generally more fertile than soils in the felsic or mixed sediment region, SPR was lower and decreased more strongly with depth in mafic soils (75% decrease in deep subsoil compared to topsoil) than in felsic soils (33% decrease) (Fig. 1a). We argue that SOC stocks in the mafic region are higher and SPR lower due to the presence of mineral related stabilization mechanisms that are lacking in other regions, consistent with the findings of Reichenbach et al. (2021). Interestingly, our data suggests that C associated with pyrophosphate extractable oxides (organo-metallic complexes) is readily available to microbial decomposers and can contribute to respiration in a short-term experiment such as ours (Fig. A3a).

In summary, the contrasting relationship of pedogenic oxides of different origin and formation to SPR and $\Delta^{14}C$ illustrates the need to improve our understanding of metal-organic interactions and their role in C stabilization in tropical soils as our results seemingly confirm (role of metal oxides) and also contradict (role of clay) findings from younger soils in the temperate zone (Khomo et al., 2017). Our results, linked to those of Reichenbach et al. (2021), show that the presence or absence of mineral stabilization mechanisms is particularly important for long-term soil C stocks in tropical soils, varying largely with soil parent material, while short-term respiration relies on readily available C sources."

*Rev 2 Comment 13: Fig 5. Can you bold the p values for what is significant here? Same for the similar graphs in the Appendix.*

**Our response:** Thank you for this suggestion. We have revised all figures and put in bold significant correlations and their corresponding p-values, and amended the figure captions accordingly.

Rev 2 Comment 14: *502: specify high C:N here*

**Our response:** We have revised this line (464) as follow:

"While fossil organic C in our study region is of poor quality as indicated by depleted N and high C:N values (153.9 ± 68.5)"